# Monochromatic Edges in Complete Multipartite Hypergraphs

**Teeradej Kittipassorn \*** and **Boonpipop Sirirojrattana**

Department of Mathematics and Computer Science, Faculty of Science, Chulalongkorn University, Bangkok 10330, Thailand
* Correspondence: teeradej.k@chula.ac.th

**Abstract:** Consider the following problem. In a school with three classes containing $n$ students each, given that their genders are unknown, find the minimum possible number of triples of same-gender students—not all of which are from the same class. Muaengwaeng asked this question and conjectured that the minimum scenario occurs when the classes are all boys, all girls and half-and-half. In this paper, we solve many generalizations of the problem including when the school has more than three classes, when triples are replaced by groups of larger sizes, when the classes are of different sizes, and when gender is replaced by other non-binary attributes.

**Keywords:** graph; hypergraph; coloring

## 1. Introduction

A *hypergraph* is a pair $(V, E)$ where $V$ is a finite set of vertices and $E$ is a collection of subsets of $V$. Each subset in $E$ is called an *edge*. An *r-uniform* hypergraph contains only edges of size $r$, and if it contains all possible edges of size $r$, this $r$-uniform hypergraph is said to be *complete*. According to the definition, an edge of a hypergraph is a collection of vertices.

From the time that hypergraphs were introduced, many researchers have been studying hypergraphs as a generalization of graphs, and many theorems in graph theory have been extended to hypergraphs (see [1–5]). Paths and cycles of hypergraphs were also extended and studied (see [6]). Coloring, as one of the most popular topics in graph theory, was also a part of those interesting extensions (see [7,8]). This also led to a new way of studying chromatic numbers; for example, Brooks's Theorem, another famous coloring theorem, was also successfully extended to hypergraphs (see [9]).

Usually, the term 'monochromatic' is used to describe a collection of edges of the same color (see [10,11]), but here, we color the vertices and consider a monochromatic collection of vertices. Given a hypergraph, an *m-coloring* is an assignment of a color to each vertex of the hypergraph from $m$ available colors. An edge is said to be *monochromatic* if all vertices in it have the same color. Some research (see [12]) focused on *proper coloring* and *defective coloring*, which involve colorings where monochromatic edges do not exist or exist only in some limited amount (see [13,14]). Some hypergraphs might have no desired defective coloring; however, we could determine the coloring that gives the fewest monochromatic edges with a given amount of colors.

The complexity of general hypergraphs has led to studies focusing on hypergraphs that have an orderly and symmetric structure, such as $k$-partite hypergraphs. A *balanced k-partite r-uniform hypergraph* has $k$ vertex classes $V_1, V_2, \ldots, V_k$ of the same size. There are two natural generalizations of $k$-partite graphs to hypergraphs. First, each edge is an $r$-subset of $\bigcup_{i=1}^{k} V_i$, all of whose vertices are from different classes. The second definition of

edges is that each edge is an *r*-subset, not all of whose vertices are from the same class. In the paper, we will use the latter definition.

The problem in the abstract can be restated in the language of hypergraphs as follows.

**Problem 1.** *Which 2-coloring minimizes the number of monochromatic edges of a balanced complete tripartite 3-uniform hypergraph?*

This question was asked by Muaengwaeng [15]. Given that those two colors are red and blue, she conjectured that the minimum coloring occurs when those three classes are all blue, all red and half-and-half. In this paper, we solve many generalizations of Problem 1. First, we study a balanced complete *k*-partite *r*-uniform hypergraph.

**Theorem 1.** *Let $n \geqslant r \geqslant 3$ and $k \geqslant 2$. The 2-coloring minimizing the number of monochromatic edges of a balanced complete k-partite r-uniform hypergraph with n vertices in each class is as follows:*

1.  *Color all vertices of the first $\lfloor \frac{k}{2} \rfloor$ classes with red;*
2.  *Color all vertices of the last $\lfloor \frac{k}{2} \rfloor$ classes with blue;*
3.  *If there is another class, color the vertices of that class such that the number of red and blue vertices are as equal as possible.*

    *Moreover, this coloring is unique up to a permutation of colors and classes.*

The case where $r = k = 3$ was presented in a conference [16] by the second author. The key idea is to calculate the change in the number of monochromatic edges when a vertex is recolored. We use this to find the minimum coloring among those with a fixed number of red vertices. Then, we compare these minimum colorings.

The proof of Theorem 1 gives a clue on how to prove a more general case when each class does not contain the same number of vertices.

**Theorem 2.** *For any unbalanced complete tripartite 3-uniform hypergraph H with the numbers of vertices of the first, second and third classes, $n_1 \leqslant n_2 \leqslant n_3$, where $n_3 \geqslant 3$ and $n_1 + n_2 + n_3 = N$, a 2-coloring minimizing the number of monochromatic edges of H is as follows:*

1.  *If $n_1 + n_2 > n_3$, color all vertices in the second and third classes with blue and red, respectively, and color $\left\lceil \frac{N^2 - 3N - n_1^2 - 2n_1 n_3 + 3n_1 + 4n_3}{2(N - n_1)} \right\rceil - n_3$ vertices in the first class with red.*
2.  *If $n_1 + n_2 \leqslant n_3$, then color all vertices in the first and second classes with red and color the third class with blue.*

    *Moreover, each coloring is unique up to a permutation of colors unless $(n_1, n_2, n_3) = (2, n, n + 1)$ for $n \geqslant 2$ in which case there is another extremal coloring, namely a coloring such that vertices in the third class are all red, all vertices in the second class are all blue and the first class has one red and one blue vertex.*

The proof consists of two parts. First, by swapping a red vertex and a blue vertex in different classes, we conclude that a minimum coloring must be in one of the 12 canonical forms. Then, we compare the numbers of monochromatic edges between the forms.

Finally, we study the number of monochromatic edges of balanced complete *k*-partite *r*-uniform hypergraphs except, this time, up to three colors will be available. The problem of minimizing the number of monochromatic edges becomes more complicated as it is not a simple two-way comparison between red and blue. The results are divided upon the remainder of the number of vertex classes, *k*, divided by 3. We are able to solve the cases $k \equiv 0, 1 \pmod{3}$.

**Theorem 3.** *Let $n \geqslant r \geqslant 3$ and $k \geqslant 3$. For any balanced complete k-partite r-uniform hypergraph, H, with n vertices in each class, if $k \equiv 0 \pmod{3}$, the 3-coloring minimizing the number of monochromatic edges of H is as follows:*

1.  Color all vertices of the first $\frac{k}{3}$ classes with red;
2.  Color all vertices of the next $\frac{k}{3}$ classes with blue;
3.  Color all vertices of the last $\frac{k}{3}$ classes with green.

If $k \equiv 1 \pmod 3$, the 3-coloring minimizing the number of monochromatic edges of H is as follows:

1.  Color all vertices of the first $\left\lfloor \frac{k}{3} \right\rfloor$ classes with red;
2.  Color all vertices of the next $\left\lfloor \frac{k}{3} \right\rfloor$ classes with blue;
3.  Color all vertices of the next $\left\lfloor \frac{k}{3} \right\rfloor$ classes with green;
4.  Color all vertices of the last class such that the number of red, blue and green vertices are as equal as possible.

Moreover, each coloring is unique up to a permutation of colors.

We use a similar idea to prove Theorem 3, but we need to develop some new lemmas to construct the canonical forms of the colorings.

The rest of this paper is organized as follows. In Section 2, we introduce some notations and useful properties that will be used throughout the paper. Later, we consider some straightforward cases. Sections 3–5 are devoted to proving Theorems 1–3, respectively. Finally, we conclude in Section 6 with a discussion of some open problems.

## 2. Counting Monochromatic Edges

First, we will introduce some notations that will be used throughout the paper. Later, we will mainly discuss some useful properties of binomial coefficients and some trivial cases of the problem.

### 2.1. The Number of Monochromatic Edges of a 2-Coloring of Balanced Complete k-Partite r-Uniform Hypergraphs

Let H be a balanced complete $k$-partite an $(r+1)$-uniform hypergraph with $n$ vertices in each class. We consider $(r+1)$-uniform instead of an $r$-uniform hypergraph for simple calculation. Let $c$ be a coloring of H with $x_i$ red vertices in the $i$th class and let $X = x_1 + x_2 + \cdots + x_k$. Let $M(H, c)$ be the number of monochromatic edges of H with coloring $c$. Then,

$$M(H, c) = \left[ \binom{X}{r+1} - \sum_{i=1}^{k} \binom{x_i}{r+1} \right] + \left[ \binom{kn - X}{r+1} - \sum_{i=1}^{k} \binom{n - x_i}{r+1} \right].$$

This function is the main method to count the number of monochromatic edges. However, this function alone is not enough for comparing the numbers of monochromatic edges of all colorings. Let $\triangle_i M(H, c)$ be the change in the number of monochromatic edges when a blue vertex in the $i$th class is recolored (if possible). The change is equal to the difference between the number of monochromatic edges containing the vertex that will be recolored before and after the recoloring. Then,

$$\triangle_i M(H, c) = \left[ \binom{X}{r} - \binom{x_i}{r} \right] - \left[ \binom{kn - X - 1}{r} - \binom{n - x_i - 1}{r} \right].$$

We sometimes simply write $\triangle M(H, c)$ instead of $\triangle_i M(H, c)$ if the class of the color-changing vertex is clear.

### 2.2. The Number of Monochromatic Edges of a 2-Coloring of Unbalanced Complete Tripartite 3-Uniform Hypergraphs

Let H be an unbalanced complete tripartite 3-uniform hypergraph with the numbers of vertices of the first, second and third classes equal to $n_1 \leqslant n_2 \leqslant n_3$, respectively, and

let $N = n_1 + n_2 + n_3$. Let $c$ be the coloring of $H$ with the number of red vertices of the first, second and third classes equal to $x_1$, $x_2$ and $x_3$, respectively, and let $X = x_1 + x_2 + x_3$. Then,

$$M(H,c) = \left[ \binom{X}{3} - \sum_{i=1}^{3} \binom{x_i}{3} \right] + \left[ \binom{N-X}{3} - \sum_{i=1}^{3} \binom{n_i - x_i}{3} \right],$$

and

$$\triangle_i M(H,c) = \left[ \binom{X}{2} - \binom{x_i}{2} \right] - \left[ \binom{N-X-1}{2} - \binom{n_i - x_i - 1}{2} \right].$$

*2.3. The Number of Monochromatic Edges of a 3-Coloring of Balanced Complete k-Partite r-Uniform Hypergraphs*

Let $H$ be a balanced complete $k$-partite $(r+1)$-uniform hypergraph with $n$ vertices in each class. Let $c$ be the 3-coloring of $H$ with the numbers of red, blue and green vertices of the $i$th class equal to $r_i$, $b_i$ and $g_i$, respectively, and let $R$, $B$ and $G$ be the total numbers of red, blue and green vertices, respectively. Note that $R = \sum_{i=1}^{k} r_i$, $B = \sum_{i=1}^{k} b_i$ and $G = \sum_{i=1}^{k} g_i$. Moreover, $n = r_i + b_i + g_i$ for each $i = 1, 2, \ldots, k$ and $kn = R + B + G$. Then,

$$M(H,c) = \left[ \binom{R}{r+1} - \sum_{i=1}^{k} \binom{r_i}{r+1} \right] + \left[ \binom{B}{r+1} - \sum_{i=1}^{k} \binom{b_i}{r+1} \right] + \left[ \binom{G}{r+1} - \sum_{i=1}^{k} \binom{g_i}{r+1} \right].$$

We write $\triangle_i M(H,c)$ for the change in the number of monochromatic edges when a blue vertex in the $i$th class is recolored to red (if possible). Then,

$$\triangle_i M(H,c) = \left[ \binom{R}{r} - \binom{r_i}{r} \right] - \left[ \binom{B-1}{r} - \binom{b_i - 1}{r} \right].$$

The change can be calculated similarly for any recoloring with other color combinations.

*2.4. Properties of Binomial Coefficients*

We will additionally introduce some standard tools that will be applied throughout this paper.

**Proposition 1.** *For any non-negative integers $a, b, c$ and $d$ with $c \leqslant a \leqslant b \leqslant d$, if $a + b \leqslant c + d$, then $\binom{a}{r} + \binom{b}{r} \leqslant \binom{c}{r} + \binom{d}{r}$ for any positive integer $r$. Moreover, the equality holds if and only if $(a = c$ and $b = d)$ or $d < r$.*

Note that the inequality trivially holds when all upper indices of binomial coefficient terms are less than the lower index. This trivial condition will be found occasionally throughout our proofs of the main theorems.

**Proposition 2.** *For any non-negative integers $x_1, x_2, \ldots, x_n$ whose sum is constant and for any non-negative integer $r$, $\sum_{i=1}^{n} \binom{x_i}{r}$ is smallest if and only if $x_1, x_2, \ldots, x_n$ are as equal as possible or $\max\{x_1, x_2, \ldots, x_n\} < r$.*

**Proposition 3.** *For any non-negative integers $x_1, x_2, \ldots, x_n$ whose sum is constant, and for any non-negative integer $r$, $\sum_{i=1}^{n} \binom{x_i}{r}$ is largest if and only if all but one $x_i$ are zeros or $\sum_{i=1}^{n} x_i < r$.*

Proposition 1 is the main tool to compare binomial coefficients, while Propositions 2 and 3 are generalizations of Proposition 1, which we will apply to prove some trivial cases of the problems in the next subsections.

*2.5. Colorings of Hypergraphs with the Size of Each Class Fewer than the Size of an Edge*

In our theorems, we assume that the size of each class must be at least the size of an edge; otherwise, an edge cannot be contained in a class and our hypergraphs are just complete hypergraphs. In this subsection, we will note that the problem is trivial when $n < r$ and determine a coloring that has the minimum number of monochromatic edges. Let $H$ be a complete $r$-uniform hypergraph, and let $c$ be a coloring of $H$. Then,

$$M(H,c) = \binom{R}{r} + \binom{B}{r}$$

if $c$ is a 2-coloring with $R$ red and $B$ blue vertices, and

$$M(H,c) = \binom{R}{r} + \binom{B}{r} + \binom{G}{r}$$

if $c$ is a 3-coloring with $R$ red, $B$ blue and $G$ green vertices.

By Proposition 1, $M(H,c)$ is smallest if $R$, $B$ (and $G$) are as equal as possible. Hence, a coloring such that the numbers of vertices of each color are as equal as possible has the minimum number of monochromatic edges.

*2.6. Colorings of Hypergraphs with the Number of Classes Fewer than or Divisible by the Number of Colors*

In this subsection, we will consider colorings where the number of classes is fewer than or divisible by the number of colors and determine the colorings with the minimum number of monochromatic edges. We consider not only the colorings with 2 or 3 colors but also $m$-colorings with $m$ greater than 3.

First, we consider the hypergraphs with the number of classes fewer than the number of colors. There are no monochromatic edges of a fixed color only when all vertices of that color are contained in at most one class, or the number of vertices of that color is fewer than $r$. Hence, the colorings such that each color appears in at most one class or appears on fewer than $r$ vertices are the only colorings with no monochromatic edges. Such colorings exist when the number of classes is fewer than the number of colors.

The determination of the minimum coloring of the remaining case is straightforward from the Propositions 1 and 2.

**Proposition 4.** *Let $n \geqslant r$ and let $k$ be divisible by $m$. The $m$-coloring minimizing the number of monochromatic edges of a balanced complete $k$-partite $r$-uniform hypergraph with $n$ vertices in each class is the coloring with equal numbers of vertices of each color and no polychromatic class.*

**Proof.** Suppose that $n \geqslant r$. Let $H$ be a balanced complete $k$-partite $r$-uniform hypergraph with $n$ vertices in each class, and let $c$ be an $m$-coloring of $H$ such that $k$ is divisible by $m$. Let $x_{li}$ be the number of vertices with the $l$th color in the $i$th class, and let $X_l$ be the total number of vertices with the $l$th color. Then,

$$M(H,c) = \sum_{l=1}^{m} \left[ \binom{X_l}{r} - \sum_{i=1}^{k} \binom{x_{li}}{r} \right].$$

We will show that the coloring $c^*$ with equal numbers of vertices of each color and no polychromatic class is the minimum coloring by directly comparing the numbers of monochromatic edges of $c$ and $c^*$. By Propositions 1 and 2,

$$M(H,c) = \sum_{l=1}^{m}\left[\binom{X_l}{r} - \sum_{i=1}^{k}\binom{x_{li}}{r}\right]$$

$$\geqslant m\binom{\frac{1}{m}\sum_{l=1}^{m}X_l}{r} - \sum_{i=1}^{k}\binom{\sum_{l=1}^{m}x_{li}}{r}$$

$$= m\binom{\frac{kn}{m}}{r} - \sum_{i=1}^{k}\binom{n}{r} = M(H,c^*).$$

The equality only holds when ($X_i = \frac{kn}{m}$ for each $i$ and each class is monochromatic) or $n < r$, but the latter is impossible. Hence, $c^*$ is the unique coloring with the minimum number of monochromatic edges. □

The proof in this subsection is a straightforward comparison due to the simplicity of color distribution. However, in other general cases, they are much more complicated.

### 3. Proof of Theorem 1

Let $H$ be a balanced complete $k$-partite $(r+1)$-uniform hypergraph with $n \geqslant r+1$ vertices in each class where $k \geqslant 2$ and $r \geqslant 2$. Let $c$ be a coloring of $H$ with $x_i$ red vertices in the $i$th class, and let $X = x_1 + x_2 + \cdots + x_k$. We may assume that $X \leqslant \lfloor \frac{kn}{2} \rfloor$; otherwise, we relabel the names of the colors. Note that if $k$ is even, the proof is completed by Proposition 3. However, we will not need to assume that $k$ is odd in the following proof.

We will calculate $M(H,c)$ in a new manner by summing each change when a blue vertex is recolored into red one by one starting from the all-blue hypergraph until we reach $c$. Let $c_0$ be the all-blue coloring, and let $c_j$ be the coloring after the $j$th change in $H$. Thus,

$$M(H,c) = M(H,c_0) + \sum_{j=0}^{X-1}\triangle M(H,c_j) = \binom{kn}{r+1} - k\binom{n}{r+1} + \sum_{j=0}^{X-1}\triangle M(H,c_j).$$

We suppose that the vertices in the first class of the all-blue hypergraph will be recolored first to match the first class of $c$ and then continue to the next class. Note that $c_j$ has $j$ red vertices. Let the $i$th class be the class containing the blue vertex that will be recolored, and let $x$ be the number of red vertices in that class. Then, from Section 2.1,

$$\triangle M(H,c_j) = \left[\binom{j}{r} - \binom{x}{r}\right] - \left[\binom{kn-j-1}{r} - \binom{n-x-1}{r}\right]$$

$$= \left[\binom{j}{r} - \binom{kn-j-1}{r}\right] - \left[\binom{x}{r} - \binom{n-x-1}{r}\right].$$

Note that while each vertex in the changing class is being recolored, the term $x$ ascends from 0 to $x_i - 1$. Thus,

$$M(H,c) = \binom{kn}{r+1} - k\binom{n}{r+1} + \sum_{j=0}^{X-1}\triangle M(H,c_j)$$

$$= \binom{kn}{r+1} - k\binom{n}{r+1} + \sum_{j=0}^{X-1}\left[\binom{j}{r} - \binom{kn-j-1}{r}\right] - \sum_{i=1}^{k}\sum_{x=0}^{x_i-1}\left[\binom{x}{r} - \binom{n-x-1}{r}\right].$$

In this way, if we consider only the colorings with $X$ red vertices, then the terms

$$\binom{kn}{r+1} - k\binom{n}{r+1} + \sum_{j=0}^{X-1}\left[\binom{j}{r} - \binom{kn-j-1}{r}\right]$$

in the function $M(H,c)$ are constant. Only the term

$$\sum_{i=1}^{k} \sum_{x=0}^{x_i-1} \left[ \binom{x}{r} - \binom{n-x-1}{r} \right]$$

is distinct, and we denote this term by $S(x_1, x_2, \ldots, x_k)$. Hence, the coloring with a maximum value of $S(x_1, x_2, \ldots, x_k)$ will have the minimum number of monochromatic edges.

**Claim 1.** *Among the colorings with a constant total number $X$ of red vertices, the coloring $c_X^*$ with the minimum number of polychromatic classes has the minimum number of monochromatic edges. Moreover, the minimum coloring is unique up to a permutation of classes.*

**Proof.** Consider a coloring with the number of red vertices in the $i$th class equal to $x_i$, where $x_1 + x_2 + \cdots + x_k = X$ and $x_1 \geqslant x_2 \geqslant \cdots \geqslant x_k$. Suppose that the coloring is not $c_X^*$. Therefore, there exist classes $l < m$ such that $x_l \neq n$ and $x_m \neq 0$. Next, we will compare the terms

$$S(x_1, \ldots, x_l, \ldots, x_m, \ldots, x_k)$$

and

$$S(x_1, \ldots, x_l + 1, \ldots, x_m - 1, \ldots, x_k).$$

which corresponds to swapping a red vertex from the $m$th class with a blue vertex from the $l$th class. Thus, since $x_l \geqslant x_m$,

$$
\begin{aligned}
&S(x_1, \ldots, x_l + 1, \ldots, x_m - 1, \ldots, x_k) - S(x_1, \ldots, x_l, \ldots, x_m, \ldots, x_k) \\
&= \sum_{x=0}^{x_l+1-1} \left[ \binom{x}{r} - \binom{n-x-1}{r} \right] + \sum_{x=0}^{x_m-1-1} \left[ \binom{x}{r} - \binom{n-x-1}{r} \right] \\
&\quad - \sum_{x=0}^{x_l-1} \left[ \binom{x}{r} - \binom{n-x-1}{r} \right] - \sum_{x=0}^{x_m-1} \left[ \binom{x}{r} - \binom{n-x-1}{r} \right] \\
&= \left[ \binom{x_l}{r} - \binom{x_m-1}{r} \right] + \left[ \binom{n-x_m}{r} - \binom{n-x_l-1}{r} \right] \geqslant 0.
\end{aligned}
$$

The equality only holds when all upper indices of the binomial coefficient terms are less than $r$. The swapping resulted in fewer or equal monochromatic edges. We will continue swapping as long as possible to reduce the number of polychromatic classes. The inequality is strict at some point since either $x_l$ will eventually equal $n - 1$, in which case $x_l = n - 1 \geqslant r$, or $x_m$ will eventually equal 1, in which case $n - x_m = n - 1 \geqslant r$. This implies that the original coloring has strictly more monochromatic edges than some coloring. Hence, $c_X^*$ is the unique coloring with the minimum number of monochromatic edges among those with $X$ red vertices. $\square$

We have determined the minimum coloring for each value of $X$. Next, we will make comparisons between colorings with different values of $X$. We will show that $M(H, c_X^*) > M(H, c_{X+1}^*)$ for all $X \leqslant \lfloor \frac{kn}{2} \rfloor - 1$. Let $c_X^*$ be the minimum coloring with $X \leqslant \lfloor \frac{kn}{2} \rfloor - 1$ red vertices. Suppose that the polychromatic class of $c_X^*$ is the $i$th class, but if $c_X^*$ has no polychromatic class, suppose the $i$th class is an all blue class. Observe that

$$
\begin{aligned}
M(H, c_{X+1}^*) - M(H, c_X^*) &= \triangle_i M(H, c_X^*) \\
&= \left[ \binom{X}{r} - \binom{x_i}{r} \right] - \left[ \binom{kn - X - 1}{r} - \binom{n - x_i - 1}{r} \right] \\
&= \left[ \binom{X}{r} + \binom{n - x_i - 1}{r} \right] - \left[ \binom{kn - X - 1}{r} + \binom{x_i}{r} \right].
\end{aligned}
$$

The following proof will be divided into two cases according to the value of $x_i$.

Case 1: $\frac{n}{2} \leqslant x_i < n$.

Thus,

$$X - (kn - X - 1) = 2\left(X + \frac{1}{2} - \frac{kn}{2}\right) \leqslant 2\left(\left\lfloor \frac{kn}{2} \right\rfloor - \frac{kn}{2} - \frac{1}{2}\right) < 0,$$

and

$$(n - x_i - 1) - x_i = 2\left(\frac{n}{2} - x_i - \frac{1}{2}\right) < 0.$$

Hence, $\triangle_i M(H, c_X^*) \leqslant 0$, but $\triangle_i M(H, c_X^*) \neq 0$ since one of the upper indices is at least $r$. Indeed, $kn - X - 1 \geqslant \left\lceil \frac{kn}{2} \right\rceil \geqslant n > r$ because $k \geqslant 2$.

Case 2: $0 \leqslant x_i < \frac{n}{2}$.

We will show that $\triangle_i M(H, c_X^*) < 0$ by Proposition 1. As in Case 1,

$$X - (kn - X - 1) < 0.$$

Moreover,

$$(n - x_i - 1) - x_i = 2\left(\frac{n}{2} - x_i - \frac{1}{2}\right) \geqslant 0.$$

Suppose that there are $k^*$ red classes in $c_X^*$, i.e., $X = k^*n + x_i$. Since $X \leqslant \lfloor \frac{kn}{2} \rfloor - 1$, we have $k^* \leqslant \frac{k-1}{2}$. Thus,

$$(X + n - x_i - 1) - (x_i + kn - X - 1) = 2\left(k^*n - \frac{k-1}{2}n\right) \leqslant 0.$$

Similarly, $kn - X - 1 > r$. Hence, by Proposition 1, $\triangle_i M(H, c_X^*) < 0$.

By the two cases, $c_X^*$ contains strictly more monochromatic edges than $c_{X+1}^*$ does, given that $X \leqslant \lfloor \frac{kn}{2} \rfloor - 1$. Consequently, we can conclude that the unique coloring with the minimum number of monochromatic edges among all minimum colorings $c_X^*$ is $c_{\lfloor \frac{kn}{2} \rfloor}^*$.

Together with the claim, $c_{\lfloor \frac{kn}{2} \rfloor}^*$ is the unique coloring with the minimum number of monochromatic edges.

Note that, in the claim, we determine $c_X^*$ by means of determining the coloring with maximum $S(x_1, x_2, \ldots, x_k)$. On the contrary, we could determine the coloring with a constant number $X$ of red vertices that has the maximum number of monochromatic edges by showing conversely that the coloring with minimum $S(x_1, x_2, \ldots, x_k)$ is the coloring such that $x_1, x_2, \ldots, x_k$ are as equal as possible. However, this is out of our topic.

## 4. Proof of Theorem 2

Let $H$ be an unbalanced complete tripartite 3-uniform hypergraph with the numbers of vertices of the first, second and third classes $n_1 \leqslant n_2 \leqslant n_3$, and let $N = n_1 + n_2 + n_3$. Let $c$ be the coloring of $H$ with the number of red vertices of the first, second and third classes equal to $x_1, x_2$ and $x_3$, respectively, and let $X = x_1 + x_2 + x_3$.

We divide the proof into two subsections according to the size of the smallest class. In the first subsection, a similar idea as in the proof of Theorem 1 is extended to determine the minimum coloring when the number of vertices of each class is at least 3. The second subsection is mainly about hypergraphs with some small classes.

### 4.1. Hypergraphs with $n_1 \geqslant 3$

Assume that $n_1 \geqslant 3$. Let $\triangle_{ii'} M(H, c)$ be the change in the number of monochromatic edges if a blue vertex in the $i$th class is recolored into red and a red vertex in the $i'$th class is recolored into blue. The process will be called *swapping*, which results in a new coloring, say

$c'$. We compute $\triangle_{ii'} M(H, c)$ by comparing the number of monochromatic edges containing those vertices undergone swapping before and after the swapping process. Thus,

$$
\begin{aligned}
\triangle_{ii'} M(H, c) &= \left[ \binom{x_1 + x_2 + x_3 - 1}{2} - \binom{x_i}{2} + \binom{N - x_1 - x_2 - x_3 - 1}{2} - \binom{n_{i'} - x_{i'}}{2} \right] \\
&\quad - \left[ \binom{x_1 + x_2 + x_3 - 1}{2} - \binom{x_{i'} - 1}{2} + \binom{N - x_1 - x_2 - x_3 - 1}{2} - \binom{n_i - x_i - 1}{2} \right] \\
&= \left[ \binom{x_{i'} - 1}{2} + \binom{n_i - x_i - 1}{2} \right] - \left[ \binom{x_i}{2} + \binom{n_{i'} - x_{i'}}{2} \right].
\end{aligned}
$$

A *successful swapping* is a swapping in such a way that the number of monochromatic edges is reduced, i.e., $\triangle_{ii'} M(H, c) < 0$.

**Lemma 1.** *If* $\triangle_{ii'} M(H, c) \leqslant 0$, *then* $\triangle_{ii'} M(H, c') < 0$.

**Proof.** Observe that

$$
\begin{aligned}
\triangle_{ii'} M(H, c') &= \left[ \binom{(x_{i'} - 1) - 1}{2} + \binom{n_i - (x_i + 1) - 1}{2} \right] - \left[ \binom{x_i + 1}{2} + \binom{n_{i'} - (x_{i'} - 1)}{2} \right] \\
&\leqslant \left[ \binom{x_{i'} - 1}{2} + \binom{n_i - x_i - 1}{2} \right] - \left[ \binom{x_i}{2} + \binom{n_{i'} - x_{i'}}{2} \right] = \triangle_{ii'} M(H, c) \leqslant 0.
\end{aligned}
$$

The equality holds only when $(x_{i'} - 1 < 2$ and $n_i - x_i - 1 < 2)$ and $(x_i + 1 < 2$ and $n_{i'} - x_{i'} + 1 < 2)$ and $\triangle_{ii'} M(H, c) = 0$. Suppose that $\triangle_{ii'} M(H, c') = 0$. We have that $x_i = 0$ and $x_{i'} < 3$. Thus, $n_i = n_i - x_i < 3$ and $n_{i'} - 2 \leqslant n_{i'} - x_{i'} < 1$, i.e., $n_{i'} < 3$, which contradicts $3 \leqslant n_1 \leqslant n_2 \leqslant n_3$. $\square$

Lemma 1 means that if a swapping can be carried out without increasing the number of monochromatic edges, another swapping in the same direction will be successful (if there are red and blue vertices to be swapped). The process of successful swappings will terminate when one of the two classes (or both) is monochromatic.

**Lemma 2.** *If* $\triangle_{ii'} M(H, c) \geqslant 0$, *then* $\triangle_{i'i} M(H, c) < 0$.

**Proof.** Observe that

$$
\begin{aligned}
\triangle_{i'i} M(H, c) &= \left[ \binom{x_i - 1}{2} + \binom{n_{i'} - x_{i'} - 1}{2} \right] - \left[ \binom{x_{i'}}{2} + \binom{n_i - x_i}{2} \right] \\
&\leqslant \left[ \binom{x_i}{2} + \binom{n_{i'} - x_{i'}}{2} \right] - \left[ \binom{x_{i'} - 1}{2} + \binom{n_i - x_i - 1}{2} \right] \\
&= - \triangle_{ii'} M(H, c) \leqslant 0.
\end{aligned}
$$

The equality holds only when $(x_i < 2$ and $n_{i'} - x_{i'} < 2)$ and $(x_{i'} < 2$ and $n_i - x_i < 2)$ and $\triangle_{ii'} M(H, c) = 0$. Suppose that $\triangle_{i'i} M(H, c) = 0$. We have that $x_i \leqslant 1$ and $x_{i'} \leqslant 1$. Thus, $n_i - 1 \leqslant = n_i - x_i < 2$ and $n_{i'} - 1 \leqslant n_{i'} - x_{i'} < 2$, i.e., $n_i < 3$ and $n_{i'} < 3$, which contradicts $3 \leqslant n_1 \leqslant n_2 \leqslant n_3$. $\square$

Note that, for any coloring $c$, if $c$ contains two classes, the $i$th and $i'$th, which are polychromatic, then a swapping can be carried out in two directions as follows.

1.  Swapping a red vertex of the $i$th class with a blue vertex of the $i'$th class.
2.  Swapping a blue vertex of the $i$th class with a red vertex of the $i'$th class.

By Lemma 2, one of the two directions is successful. Moreover, by Lemma 1, we can continue swapping in the same direction until one of the two classes is monochromatic and has fewer monochromatic edges. Hence, the coloring with the minimum number of monochromatic edges among colorings with a constant number of red vertices must have

at most one polychromatic class. We will list all these forms in the following Table 1, which will be the candidates for the coloring with the minimum number of monochromatic edges.

**Table 1.** The twelve canonical forms.

| Canonical Forms | 1st Class | 2nd Class | 3rd Class |
|:---:|:---:|:---:|:---:|
| $F_1$ | polychromatic | blue | blue |
| $F_2$ | blue | polychromatic | blue |
| $F_3$ | blue | blue | polychromatic |
| $F_4$ | red | polychromatic | blue |
| $F_5$ | red | blue | polychromatic |
| $F_6$ | polychromatic | red | blue |
| $F_7$ | blue | red | polychromatic |
| $F_8$ | polychromatic | blue | red |
| $F_9$ | blue | polychromatic | red |
| $F_{10}$ | red | red | polychromatic |
| $F_{11}$ | red | polychromatic | red |
| $F_{12}$ | polychromatic | red | red |

The first column illustrates the list of 12 canonical forms, and the remaining columns describe the colors of vertices in those classes. The terms *red* and *blue* mean all vertices in those classes are monochromatic of red and blue, respectively. On the other hand, *polychromatic* means that this class is allowed to be polychromatic, but it may be monochromatic. Note that a coloring can be considered in several canonical forms; for example, the all-blue coloring is of the form $F_1$, $F_2$ or $F_3$.

We may assume that $X \leqslant \lfloor \frac{N}{2} \rfloor$. Consequently, both $F_{11}$ and $F_{12}$ are out of our interest since the total numbers of red vertices, which are $n_1 + x_2 + n_3$ and $x_1 + n_2 + n_3$, respectively, exceed $\lfloor \frac{N}{2} \rfloor$, as shown:

$$n_1 + x_2 + n_3 = \frac{n_1 + n_3 + 2x_2}{2} + \frac{n_1 + n_3}{2} > \frac{n_2}{2} + \frac{n_1 + n_3}{2} \geqslant \left\lfloor \frac{n_1 + n_2 + n_3}{2} \right\rfloor$$

and

$$x_1 + n_2 + n_3 = \frac{n_2 + n_3 + 2x_1}{2} + \frac{n_2 + n_3}{2} > \frac{n_1}{2} + \frac{n_2 + n_3}{2} \geqslant \left\lfloor \frac{n_1 + n_2 + n_3}{2} \right\rfloor.$$

Next, we will focus on the possibility of $F_{10}$. If $c$ has total red vertices to be $X = n_1 + n_2 + x_3 \leqslant \lfloor \frac{n_1 + n_2 + n_3}{2} \rfloor$. Then,

$$n_1 + n_2 \leqslant 2\left(\left\lfloor \frac{n_1 + n_2 + n_3}{2} \right\rfloor - \frac{n_1 + n_2}{2}\right) \leqslant 2\left(\frac{n_1 + n_2 + n_3}{2} - \frac{n_1 + n_2}{2}\right) = n_3.$$

The necessary condition of a coloring $c$ of a hypergraph $H$ to be in the form $F_{10}$ is that $n_1 + n_2 \leqslant n_3$. Note that the condition $n_1 + n_2 \leqslant n_3$ is equivalent to $n_1 + n_2 \leqslant \lfloor \frac{n_1 + n_2 + n_3}{2} \rfloor \leqslant n_3$, and we call a hypergraph with this condition *type A*. On the other hand, the condition $n_1 + n_2 > n_3$ is equivalent to $n_3 \leqslant \lfloor \frac{n_1 + n_2 + n_3}{2} \rfloor < n_1 + n_2$, and we call a hypergraph with this condition *type B*. Next, we will determine the minimum coloring among those colorings with a constant number $X$ of red vertices or $c_X^*$ from the candidates $F_1$ to $F_9$, and $F_{10}$ will be considered only when $H$ is a type A hypergraph.

As in the proof of Theorem 1, we calculate $M(H, c)$ by summing each change when a blue vertex is recolored into red one by one starting from the all-blue hypergraph until we reach $c$. Let $c_0$ be the all-blue coloring and let $c_j$ be the coloring after the $j$th change in $H$. Thus,

$$M(H,c) = M(H,c_0) + \sum_{j=0}^{X-1} \triangle M(H,c_j) = \binom{N}{3} - \sum_{i=1}^{3} \binom{n_i}{3} + \sum_{j=0}^{X-1} \triangle M(H,c_j).$$

We suppose that the vertices in the first class of the all-blue hypergraph will be recolored first to match the first class of $c$ and then continue to the next class. Note that $c_j$ has $j$ red vertices. Let the $i$th class be the class containing the blue vertex that will be recolored and $x$ be the number of red vertices in that class. Then, from Section 2.2,

$$\triangle M(H,c_j) = \left[ \binom{j}{2} - \binom{x}{2} \right] - \left[ \binom{N-j-1}{r} - \binom{n_i-x-1}{2} \right]$$

$$= \left[ \binom{j}{2} - \binom{N-j-1}{2} \right] - \left[ \binom{x}{2} - \binom{n_i-x-1}{2} \right].$$

Note that while each vertex in the changing class is being recolored, the term $x$ ascends from 0 to $x_i - 1$. Thus,

$$M(H,c) = \binom{N}{3} - \sum_{i=1}^{3} \binom{n_i}{3} + \sum_{j=0}^{X-1} \triangle M(H,c_j)$$

$$= \binom{N}{3} - \sum_{i=1}^{3} \binom{n_i}{3} + \sum_{j=0}^{X-1} \left[ \binom{j}{2} - \binom{N-j-1}{2} \right] - \sum_{i=1}^{3} \sum_{x=0}^{x_i-1} \left[ \binom{x}{2} - \binom{n_i-x-1}{2} \right].$$

Similarly, if we consider only the coloring with $X$ red vertices, then the terms

$$\binom{N}{3} - \sum_{i=1}^{3} \binom{n_i}{3} + \sum_{j=0}^{X-1} \left[ \binom{j}{2} - \binom{N-j-1}{2} \right]$$

in the function $M(H,c)$ are constant. Only the term

$$\sum_{i=1}^{3} \sum_{x=0}^{x_i-1} \left[ \binom{x}{2} - \binom{n_i-x-1}{2} \right]$$

is distinct and we denote this term by $S(x_1, x_2, x_3)$. Hence, the coloring with maximum value of $S(x_1, x_2, x_3)$ will have the minimum number of monochromatic edges. Remark that if $x_i = n_i$, then the term $\sum_{x=0}^{x_i-1} \left[ \binom{x}{2} - \binom{n_i-x-1}{2} \right]$ cancels itself out. Hence, $S(n_1, x_2, x_3) = S(0, x_2, x_3)$ and similarly when $x_2 = n_2$ or $x_3 = n_3$.

Next, we will determine $c_X^*$ by considering and comparing only among possible canonical forms according to the value $X$ and type of $H$. We will divide into several cases.

Case 1: $0 \leqslant X < n_1$ (See Table 2).

**Table 2.** Possible canonical forms for Case 1.

| Possible Forms | $x_1$ | $x_2$ | $x_3$ | $S(x_1, x_2, x_3)$ |
|---|---|---|---|---|
| $F_1$ | $X$ | 0 | 0 | $S(X, 0, 0)$ |
| $F_2$ | 0 | $X$ | 0 | $S(0, X, 0)$ |
| $F_3$ | 0 | 0 | $X$ | $S(0, 0, X)$ |

We will compare among colorings in the forms $F_1$, $F_2$ and $F_3$. Note that if $n_1 = n_2$, $F_1$ and $F_2$ are the same. Similarly, if $n_1 = n_2 = n_3$, $F_1$, $F_2$ and $F_3$ are the same. Then,

$$S(0, X, 0) = \sum_{j=0}^{X-1}\left[\binom{j}{2} - \binom{n_2-j-1}{2}\right] \leqslant \sum_{j=0}^{X-1}\left[\binom{j}{2} - \binom{n_1-j-1}{2}\right] = S(X, 0, 0).$$

The equality holds only when $n_1 = n_2$ or $n_2 - 1 < 2$. Since $3 \leqslant n_2 \leqslant n_3$, we can conclude that a coloring in the form $F_1$ has fewer monochromatic edges than $F_2$ and similarly for $F_3$. Hence, $c_X^*$ is in the form $F_1$.

Case 2: $n_1 \leqslant X < \frac{n_1 + n_2}{2}$ (See Table 3).

**Table 3.** Possible canonical forms for Case 2.

| Possible Forms | $x_1$ | $x_2$ | $x_3$ | $S(x_1, x_2, x_3)$ |
|---|---|---|---|---|
| $F_2$ | $0$ | $X$ | $0$ | $S(0, X, 0)$ |
| $F_3$ | $0$ | $0$ | $X$ | $S(0, 0, X)$ |
| $F_4$ | $n_1$ | $X - n_1$ | $0$ | $S(n_1, X - n_1, 0)$ |
| $F_5$ | $n_1$ | $0$ | $X - n_1$ | $S(n_1, 0, X - n_1)$ |

In this case, we must have that $n_1 < n_2$. Similarly to Case 1, we have that $F_2$ has fewer monochromatic edges than $F_3$. Next, we will compare between colorings in the forms $F_4$ and $F_5$. If $n_2 = n_3$, then both forms are the same. Thus,

$$S(n_1, 0, X - n_1) = S(0, 0, X - n_1) \leqslant S(0, X - n_1, 0) = S(n_1, X - n_1, 0).$$

The equality holds only when $n_2 = n_3$ or $n_3 - 1 < 2$. Since $3 \leqslant n_3$, we have that $F_4$ has fewer monochromatic edges than $F_5$. Finally, we will compare between colorings in the forms $F_2$ and $F_4$. Note that $X - n_1 < \frac{n_1 + n_2}{2} - n_1 = n_2 - \frac{n_1 + n_2}{2} < n_2 - X$. Then,

$$\begin{aligned}
S(0, X, 0) &= \sum_{j=0}^{X-1}\left[\binom{j}{2} - \binom{n_2-j-1}{2}\right] \\
&= \sum_{j=0}^{(X-n_1)-1}\left[\binom{j}{2} - \binom{n_2-j-1}{2}\right] + \left[\binom{X-n_1}{2} + \binom{X-n_1+1}{2} + \cdots + \binom{X-1}{2}\right] \\
&\quad - \left[\binom{n_2-X}{2} + \binom{n_2-X+1}{2} + \cdots + \binom{n_1+n_2-X-1}{2}\right] \\
&\leqslant \sum_{j=0}^{(X-n_1)-1}\left[\binom{j}{2} - \binom{n_2-j-1}{2}\right] = S(0, X - n_1, 0) = S(n_1, X - n_1, 0).
\end{aligned}$$

The equality holds only when $n_1 + n_2 - X - 1 < 2$. Since $X < \frac{n_1 + n_2}{2}$, this occurs only when $n_1 + n_2 \leqslant 3$, i.e., $n_1 \leqslant n_2 < 3$. Since $3 \leqslant n_2$, we have that $F_4$ gives fewer monochromatic edges than $F_2$, and $c_X^*$ is in the form $F_4$.

Case 3: $\frac{n_1 + n_2}{2} \leqslant X < n_2$ (See Table 4).

**Table 4.** Possible canonical forms for Case 3.

| Possible Forms | $x_1$ | $x_2$ | $x_3$ | $S(x_1, x_2, x_3)$ |
|---|---|---|---|---|
| $F_2$ | $0$ | $X$ | $0$ | $S(0, X, 0)$ |
| $F_3$ | $0$ | $0$ | $X$ | $S(0, 0, X)$ |
| $F_4$ | $n_1$ | $X - n_1$ | $0$ | $S(n_1, X - n_1, 0)$ |
| $F_5$ | $n_1$ | $0$ | $X - n_1$ | $S(n_1, 0, X - n_1)$ |

Similarly, we must have $n_1 < n_2$ in this case. The comparisons between $F_2$ and $F_3$ and between $F_4$ and $F_5$ are similar to the previous case. Note that $X - n_1 \geqslant \frac{n_1 + n_2}{2} - n_1 =$

$n_2 - \frac{n_1 + n_2}{2} \geqslant n_2 - X$. Next, we will show the comparison between $F_2$ and $F_4$, which gives a contrary result as shown:

$$
\begin{aligned}
S(0, X, 0) &= \sum_{j=0}^{X-1} \left[ \binom{j}{2} - \binom{n_2 - j - 1}{2} \right] \\
&= \sum_{j=0}^{(X-n_1)-1} \left[ \binom{j}{2} - \binom{n_2 - j - 1}{2} \right] + \left[ \binom{X - n_1}{2} + \binom{X - n_1 + 1}{2} + \cdots + \binom{X - 1}{2} \right] \\
&\quad - \left[ \binom{n_2 - X}{2} + \binom{n_2 - X + 1}{2} + \cdots + \binom{n_1 + n_2 - X - 1}{2} \right] \\
&\geqslant \sum_{j=0}^{(X-n_1)-1} \left[ \binom{j}{2} - \binom{n_2 - j - 1}{2} \right] = S(0, X - n_1, 0) = S(n_1, X - n_1, 0).
\end{aligned}
$$

The equality holds only when $X = \frac{n_1 + n_2}{2}$ or $X - 1 < 2$. Since $\frac{n_1 + n_2}{2} \leqslant X < n_2$ and $n_1 < n_2$, the condition that $X < 3$ occurs only when $n_1 = 1$ and $n_2 = 2$ or 3. Since $3 \leqslant n_1$, we have that $F_2$ gives fewer monochromatic edges than $F_4$, and $c_X^*$ is in the form $F_2$ when $X > \frac{n_1 + n_2}{2}$. If $X = \frac{n_1 + n_2}{2}$; both forms give the same number of monochromatic edges.

From now on, the cases will be divided by whether the hypergraph is of type A or B. Case 4A: $n_2 \leqslant X < n_1 + n_2$ and $H$ is a type A hypergraph (See Table 5).

**Table 5.** Possible canonical forms for Case 4A.

| Possible Forms | $x_1$ | $x_2$ | $x_3$ | $S(x_1, x_2, x_3)$ |
|:---:|:---:|:---:|:---:|:---:|
| $F_3$ | 0 | 0 | $X$ | $S(0, 0, X)$ |
| $F_4$ | $n_1$ | $X - n_1$ | 0 | $S(n_1, X - n_1, 0)$ |
| $F_5$ | $n_1$ | 0 | $X - n_1$ | $S(n_1, 0, X - n_1)$ |
| $F_6$ | $X - n_2$ | $n_2$ | 0 | $S(X - n_2, n_2, 0)$ |
| $F_7$ | 0 | $n_2$ | $X - n_2$ | $S(0, n_2, X - n_2)$ |

Similarly to Case 2, we have that $F_4$ has fewer monochromatic edges than $F_5$. Next, we will compare between colorings in the forms $F_6$ and $F_7$. If $n_1 = n_2 = n_3$, then the forms $F_4, F_5, F_6$ and $F_7$ are the same. Thus,

$$
S(0, n_2, X - n_2) = S(0, 0, X - n_2) \leqslant S(X - n_2, 0, 0) = S(X - n_2, n_2, 0).
$$

The equality holds only when $n_1 = n_3$ or $n_3 - 1 < 2$. Since $n_3 \geqslant 3$, we have that $F_6$ has fewer monochromatic edges than $F_7$. Next, we will compare between colorings in the forms $F_4$ and $F_6$. If $n_1 = n_2$, then both forms are the same. Suppose that $n_1 < n_2$. Thus, since $X < n_1 + n_2$,

$$
\begin{aligned}
S(n_1, X - n_1, 0) &= S(0, X - n_1, 0) \\
&= \left[ \binom{0}{2} + \binom{1}{2} + \cdots + \binom{X - n_2 - 1}{2} \right] \\
&\quad - \left[ \binom{n_1 + n_2 - X}{2} + \binom{n_1 + n_2 - X + 1}{2} + \cdots + \binom{n_1 - 1}{2} \right] \\
&\quad + \left[ \binom{X - n_2}{2} + \binom{X - n_2 + 1}{2} + \cdots + \binom{X - n_1 - 1}{2} \right] \\
&\quad - \left[ \binom{n_1}{2} + \binom{n_1 + 1}{2} + \cdots + \binom{n_2 - 1}{2} \right]
\end{aligned}
$$

$$\leqslant \left[ \binom{0}{2} + \binom{1}{2} + \cdots + \binom{X - n_2 - 1}{2} \right]$$
$$- \left[ \binom{n_1 + n_2 - X}{2} + \binom{n_1 + n_2 - X + 1}{2} + \cdots + \binom{n_1 - 1}{2} \right]$$
$$= S(X - n_2, 0, 0) = S(X - n_2, n_2, 0).$$

The equality only holds when $n_2 - 1 < 2$. Since $n_2 \geqslant 3$, we have that $F_6$ has fewer monochromatic edges than $F_4$. Finally, we will compare between colorings in the forms $F_3$ and $F_6$. Then, since $n_2 \leqslant X < n_1 + n_2 \leqslant n_3$,

$$S(0, 0, X) = \left[ \binom{0}{2} + \binom{1}{2} + \cdots + \binom{X - 1}{2} \right] - \left[ \binom{n_3 - X}{2} + \binom{n_3 - X + 1}{2} + \cdots + \binom{n_3 - 1}{2} \right]$$
$$\leqslant \left[ \binom{0}{2} + \binom{1}{2} + \cdots + \binom{X - 1}{2} \right] - \left[ \binom{n_3 - X}{2} + \binom{n_3 - X + 1}{2} + \cdots + \binom{n_3 - 1}{2} \right]$$
$$- \left[ \binom{X - n_2}{2} + \binom{X - n_2 + 1}{2} + \cdots + \binom{X - 1}{2} \right]$$
$$+ \left[ \binom{n_3 - n_2}{2} + \binom{n_3 + n_2 + 1}{2} + \cdots + \binom{n_3 - 1}{2} \right]$$
$$- \left[ \binom{n_1 + n_2 - X}{2} + \binom{n_1 + n_2 - X + 1}{2} + \cdots + \binom{n_3 - X - 1}{2} \right]$$
$$+ \left[ \binom{n_1}{2} + \binom{n_1 + 1}{2} + \cdots + \binom{n_3 - n_2 - 1}{2} \right]$$
$$= \left[ \binom{0}{2} + \binom{1}{2} + \cdots + \binom{X - n_2 - 1}{2} \right]$$
$$- \left[ \binom{n_1 + n_2 - X}{2} + \binom{n_1 + n_2 - X + 1}{2} + \cdots + \binom{n_1 - 1}{2} \right]$$
$$= \sum_{j=0}^{(X - n_2) - 1} \left[ \binom{j}{2} - \binom{n_1 - j - 1}{2} \right] = S(X - n_2, 0, 0) = S(X - n_2, n_2, 0).$$

The equality holds only when $n_3 - 1 < 2$. Since $n_3 \geqslant 3$, $F_6$ gives fewer monochromatic edges than $F_3$, and $c_X^*$ is in the form $F_6$.

Case 5A: $n_1 + n_2 \leqslant X \leqslant \lfloor \frac{N}{2} \rfloor$ and $H$ is a type A hypergraph (See Table 6).

**Table 6.** Possible canonical forms for Case 5A.

| Possible Forms | $x_1$ | $x_2$ | $x_3$ | $S(x_1, x_2, x_3)$ |
|---|---|---|---|---|
| $F_3$ | 0 | 0 | $X$ | $S(0, 0, X)$ |
| $F_{10}$ | $n_1$ | $n_2$ | $X - n_1 - n_2$ | $S(n_1, n_2, X - n_1 - n_2)$ |

Note that this is only the case that we will consider $F_{10}$. We only compare between colorings in the forms $F_3$ and $F_{10}$. Then, since $X \leqslant \lfloor \frac{n_1 + n_2 + n_3}{2} \rfloor$,

$$S(0, 0, X) = \left[ \binom{0}{2} + \binom{1}{2} + \cdots + \binom{X - 1}{2} \right] - \left[ \binom{n_3 - X}{2} + \binom{n_3 - X + 1}{2} + \cdots + \binom{n_3 - 1}{2} \right]$$
$$= \left[ \binom{0}{2} + \binom{1}{2} + \cdots + \binom{X - n_1 - n_2 - 1}{2} \right]$$
$$- \left[ \binom{n_1 + n_2 + n_3 - X}{2} + \binom{n_1 + n_2 + n_3 - X + 1}{2} + \cdots + \binom{n_3 - 1}{2} \right]$$

$$+ \left[ \binom{X - n_1 - n_2}{2} + \binom{X - n_1 - n_2 + 1}{2} + \cdots + \binom{X - 1}{2} \right]$$

$$- \left[ \binom{n_3 - X}{2} + \binom{n_3 - X + 1}{2} + \cdots + \binom{n_1 + n_2 + n_3 - X - 1}{2} \right]$$

$$\leqslant \left[ \binom{0}{2} + \binom{1}{2} + \cdots + \binom{X - n_1 - n_2 - 1}{2} \right]$$

$$- \left[ \binom{n_1 + n_2 + n_3 - X}{2} + \binom{n_1 + n_2 + n_3 - X + 1}{2} + \cdots + \binom{n_3 - 1}{2} \right]$$

$$= S(0, 0, X - n_1 - n_2) = S(n_1, n_2, X - n_1 - n_2).$$

The equality holds only when $X = \frac{N}{2}$ or $N - X - 1 < 2$. Since $X \leqslant \lfloor \frac{N}{2} \rfloor$, the condition that $N - X < 3$ occurs only when $N \leqslant 4$. This is impossible because $n_3 \geqslant 3$. If $X = \frac{N}{2}$, then both forms have the same number of monochromatic edges. However, if we relabel the names of the colors, then both forms are the same. Hence, $F_{10}$ gives fewer monochromatic edges than $F_3$, and $c_X^*$ is in the form $F_{10}$. Next, we will focus on the other cases of type B hypergraphs.

Case 4B: $n_2 \leqslant X < n_3$ and $H$ is a type B hypergraph (See Table 7).

**Table 7.** Possible canonical forms for Case 4B.

| Possible Forms | $x_1$ | $x_2$ | $x_3$ | $S(x_1, x_2, x_3)$ |
|---|---|---|---|---|
| $F_3$ | $0$ | $0$ | $X$ | $S(0, 0, X)$ |
| $F_4$ | $n_1$ | $X - n_1$ | $0$ | $S(n_1, X - n_1, 0)$ |
| $F_5$ | $n_1$ | $0$ | $X - n_1$ | $S(n_1, 0, X - n_1)$ |
| $F_6$ | $X - n_2$ | $n_2$ | $0$ | $S(X - n_2, n_2, 0)$ |
| $F_7$ | $0$ | $n_2$ | $X - n_2$ | $S(0, n_2, X - n_2)$ |

Similarly to Case 4A, $F_4$ and $F_6$ have fewer monochromatic edges than $F_5$ and $F_7$, respectively, and $F_6$ has fewer monochromatic edges than $F_4$. Next, we have to compare between $F_3$ and $F_6$ to determine which form $c_X^*$ is. Due to the complexity of the comparison, we will conclude that $c_X^*$ is in the form $F_3$ or $F_6$.

Case 5B: $n_3 \leqslant X \leqslant \lfloor \frac{n_1 + n_2 + n_3}{2} \rfloor$ and $H$ is a type B hypergraph (See Table 8).

**Table 8.** Possible canonical forms for Case 5B.

| Possible Forms | $x_1$ | $x_2$ | $x_3$ | $S(x_1, x_2, x_3)$ |
|---|---|---|---|---|
| $F_4$ | $n_1$ | $X - n_1$ | $0$ | $S(n_1, X - n_1, 0)$ |
| $F_5$ | $n_1$ | $0$ | $X - n_1$ | $S(n_1, 0, X - n_1)$ |
| $F_6$ | $X - n_2$ | $n_2$ | $0$ | $S(X - n_2, n_2, 0)$ |
| $F_7$ | $0$ | $n_2$ | $X - n_2$ | $S(0, n_2, X - n_2)$ |
| $F_8$ | $X - n_3$ | $0$ | $n_3$ | $S(X - n_3, 0, n_3)$ |
| $F_9$ | $0$ | $X - n_3$ | $n_3$ | $S(0, X - n_3, n_3)$ |

Similarly to Case 4A, $F_4$ and $F_6$ have fewer monochromatic edges than $F_5$ and $F_7$, respectively, and $F_6$ has fewer monochromatic edges than $F_4$. Next, we will compare between colorings in the forms $F_8$ and $F_9$. If $n_1 = n_2$, then both forms are the same. Thus,

$$S(0, X - n_3, n_3) = S(0, X - n_3, 0) = S(X - n_3, 0, 0) == S(X - n_3, 0, n_3).$$

The equality holds only when $n_1 = n_2$ or $n_2 - 1 < 2$. Since $3 \leqslant n_2$, we have that $F_8$ gives fewer monochromatic edges than $F_9$. Finally, we will compare between colorings in the forms $F_6$ and $F_8$. If $n_2 = n_3$, then both forms are the same. We may suppose that $n_2 < n_3$. Thus, since $X \leqslant \lfloor \frac{N}{2} \rfloor$,

$$S(X - n_2, n_2, 0) = S(X - n_2, 0, 0)$$

$$= \left[ \binom{0}{2} + \binom{1}{2} + \cdots + \binom{X - n_3 - 1}{2} \right]$$

$$+ \left[ \binom{X - n_3}{2} + \binom{X - n_3 + 1}{2} + \cdots + \binom{X - n_2 - 1}{2} \right]$$

$$- \left[ \binom{n_1 + n_2 - X}{2} + \binom{n_1 + n_2 - X + 1}{2} + \cdots + \binom{n_1 + n_3 - X - 1}{2} \right]$$

$$- \left[ \binom{n_1 + n_3 - X}{2} + \binom{n_1 + n_3 - X + 1}{2} + \cdots + \binom{n_1 - 1}{2} \right]$$

$$\leqslant \left[ \binom{0}{2} + \binom{1}{2} + \cdots + \binom{X - n_3 - 1}{2} \right]$$

$$- \left[ \binom{n_1 + n_3 - X}{2} + \binom{n_1 + n_3 - X + 1}{2} + \cdots + \binom{n_1 - 1}{2} \right]$$

$$= S(X - n_3, 0, 0) = S(X - n_3, 0, n_3).$$

The equality holds only when $X = \frac{N}{2}$ or $n_1 + n_3 - X - 1 < 2$. First, if $X = \frac{N}{2}$, then both forms have the same number of monochromatic edges. However, if we relabel the names of the colors, then both forms are the same. We will focus on the condition of $n_1 + n_3 - X < 3$, where $3 \leqslant n_1 \leqslant n_2 < n_3 \leqslant X < \frac{N}{2}$ and $n_1 + n_2 > n_3$. Suppose that $n_1 + n_3 - X < 3$.

If $N$ is even, then $3 > n_1 + n_2 - X \geqslant n_1 + n_3 - \left( \frac{N}{2} - 1 \right) = \frac{n_1 + n_3 - n_2}{2} + 1$. Thus, $n_1 + n_3 - n_2 < 4$. Since $1 \leqslant n_3 - n_2$, we have that $n_1 < 3$, which contradicts $3 \leqslant n_1$.

If $N$ is odd, then $3 > n_1 + n_2 - X \geqslant n_1 + n_3 - \left( \frac{N-1}{2} \right) = \frac{n_1 + n_3 - n_2 + 1}{2}$. Thus, $n_1 + n_3 - n_2 < 5$. Since $1 \leqslant n_3 - n_2$, we have that $n_1 < 4$. Consequently, it is only possible when $n_1 = 3$. Note that $n_2 < n_3$ and $n_1 + n_2 = 3 + n_2 > n_3$. For $N = 3 + n_2 + n_3$ to be odd, $n_2$ and $n_3$ must have the same parity. Thus, $n_3 = n_2 + 2$ and $5 > n_1 + n_3 - n_2 = 3 + n_2 + 2 - n_2 = 5$, which is a contradiction.

Now, we have that $n_1 + n_3 - X \geqslant 3$. Hence, $F_8$ gives fewer monochromatic edges than $F_6$, and $c_X^*$ is in the form $F_8$.

To sum up, we have already determined (as shown in Tables 9 and 10) the canonical form $c_X^*$ that has the minimum number of monochromatic edges for each condition of hypergraphs and range of the number of red vertices $X$.

**Table 9.** List of best canonical forms for Type A hypergraphs $n_1 + n_2 \leqslant n_3$.

| Cases | Number of Red Vertices | Canonical Form |
|---|---|---|
| 1 | $0 \leqslant X < n_1$ | $F_1$ |
| 2 | $n_1 \leqslant X < \dfrac{n_1 + n_2}{2}$ | $F_4$ |
| 3 | $X = \dfrac{n_1 + n_2}{2}$ | $F_2$ or $F_4$ |
| | $\dfrac{n_1 + n_2}{2} < X < n_2$ | $F_2$ |
| 4A | $n_2 \leqslant X < n_1 + n_2$ | $F_6$ |
| 5A | $n_1 + n_2 \leqslant X \leqslant \lfloor \frac{N}{2} \rfloor$ | $F_{10}$ |

Note that, in Case 3, $c_X^*$ is only in the form $F_2$ when $\frac{n_1 + n_2}{2} < X < n_2$. However, if $X = \frac{n_1 + n_2}{2}$, then $F_2$ and $F_4$ give the same number of monochromatic edges. Moreover, in Case 4B, we have not compared the colorings in the form $F_3$ and $F_6$. The uniqueness of $c_X^*$ of the other cases will be also considered. We can see that the inequalities in some cases are equal when the sizes of some parts are equal, which means that those canonical forms are equivalent. For example, in Case 1, if $n_2 = n_3$, $F_2$ is equivalent to $F_3$. The remaining

inequalities are equal when $X$ is equal to some certain value, such as in Case 5A and Case 5B when $X = \frac{N}{2}$; the colorings in the forms $F_3$ and $F_{10}$ are isomorphic up to a permutation of the name of colors and so are colorings in the forms $F_6$ and $F_8$ in Case 5B.

**Table 10.** List of best canonical forms for Type B hypergraphs $n_1 + n_2 > n_3$.

| Cases | Number of Red Vertices | Canonical Form |
|---|---|---|
| 1 | $0 \leqslant X < n_1$ | $F_1$ |
| 2 | $n_1 \leqslant X < \dfrac{n_1 + n_2}{2}$ | $F_4$ |
| 3 | $X = \dfrac{n_1 + n_2}{2}$ | $F_2$ or $F_4$ |
| | $\dfrac{n_1 + n_2}{2} < X < n_2$ | $F_2$ |
| 4B | $n_2 \leqslant X < n_3$ | $F_3$ or $F_6$ |
| 5B | $n_3 \leqslant X \leqslant \lfloor \frac{N}{2} \rfloor$ | $F_8$ |

Hence, apart from Case 3, where $X = \frac{n_1 + n_2}{2}$, and Case 4B, a coloring with red vertices in the form, according to the previous table, has fewer monochromatic edges than other colorings with the same amount of red vertices and the uniqueness follows.

Next, we will make comparisons between colorings with different values of $X$. We will show that any $c_X^*$ with $X \leqslant \lfloor \frac{N}{2} \rfloor - 1$ has strictly more monochromatic edges than some colorings. We, hence, would like to show that $M(H, c_{X+1}^*) - M(H, c_X^*) < 0$ for each $0 \leqslant X < \lfloor \frac{N}{2} \rfloor - 1$. Note that, if $c_X^*$ and $c_{X+1}^*$ are in the same canonical form, we will consider $\triangle M(H, c_X^*)$ instead. Again, we will divide into several cases conforming to the value of $X$ and the type of $H$.

Case 1: $0 \leqslant X < n_1$.

We have that $c_X^*$ is in the form $F_1$ with $x_1 = X$, $x_2 = 0$ and $x_3 = 0$. Then,

$$\triangle_1 M(H, c_X^*) = \left[ \binom{X}{2} - \binom{X}{2} \right] - \left[ \binom{N - X - 1}{2} - \binom{n_1 - X - 1}{2} \right] \leqslant 0.$$

The equality holds only when $N - X - 1 < 2$, which is impossible.

Case 2: $n_1 \leqslant X < \frac{n_1 + n_2}{2}$.

We have that $c_X^*$ is in the form $F_4$ with $x_1 = n_1$, $x_2 = X - n_1$ and $x_3 = 0$. Then,

$$\triangle_2 M(H, c_X^*) = \left[ \binom{X}{2} + \binom{n_1 + n_2 - X - 1}{2} \right] - \left[ \binom{X - n_1}{2} + \binom{N - X - 1}{2} \right].$$

We will show that $\triangle_2 M(H, c_X^*) < 0$ by Proposition 1. We have $X + (n_1 + n_2 - X - 1) = n_1 + n_2 - 1 \leqslant n_3 + n_2 - 1 = (X - n_1) + (N - X - 1)$, $X - n_1 < X$ and $n_1 + n_2 - X - 1 < N - X - 1$. Since $2 \leqslant N - X - 1$, we have that $\triangle_2 M(H, c_X^*) < 0$.

Case 3: $\frac{n_1 + n_2}{2} \leqslant X < n_2$.

We have that $c_X^*$ is in the form $F_2$ or $F_4$. We will show that $\triangle_2 M(H, c_X^*) < 0$ for both forms. If $X = \frac{n_1 + n_2}{2}$ and $c_X^*$ is in the form $F_4$ with $x_1 = n_1$, $x_2 = X - n_1$ and $x_3 = 0$, then $\triangle_2 M(H, c_X^*) < 0$ similarly as in Case 2.

If $\frac{n_1 + n_2}{2} \leqslant X < n_2$ and $c_X^*$ is in the form $F_2$ with $x_1 = 0$, $x_2 = X$ and $x_3 = 0$, then

$$\triangle_2 M(H, c_X^*) = \left[ \binom{X}{2} - \binom{X}{2} \right] - \left[ \binom{N - X - 1}{2} - \binom{n_2 - X - 1}{2} \right] \leqslant 0.$$

The equality holds only when $N - X - 1 < 2$, which is impossible.

Again, from this point, the cases will be divided by whether the hypergraph is of type A or B.

Case 4A: $n_2 \leqslant X < n_1 + n_2$ and $H$ is a type A hypergraph.

We have that $c_X^*$ is in the form $F_6$ with $x_1 = X - n_2$, $x_2 = n_2$ and $x_3 = 0$. Then,

$$\triangle_1 M(H, c_X^*) = \left[ \binom{X}{2} + \binom{n_1 + n_2 - X - 1}{2} \right] - \left[ \binom{X - n_2}{2} + \binom{N - X - 1}{2} \right].$$

We will show that $\triangle_1 M(H, c_X^*) < 0$ by Proposition 1. We have $X + (n_1 + n_2 - X - 1) = n_1 + n_2 - 1 \leqslant n_1 + n_3 - 1 = (X - n_2) + (N - X - 1)$, $X - n_2 < X$ and $n_1 + n_2 - X - 1 < N - X - 1$. Since $2 \leqslant N - X - 1$, we have that $\triangle_1 M(H, c_X^*) < 0$.

Case 5A: $n_1 + n_2 \leqslant X \leqslant \lfloor \frac{N}{2} \rfloor$ and $H$ is a type A hypergraph.

We have that $c_X^*$ is in the form $F_{10}$ with $x_1 = n_1$, $x_2 = n_2$ and $x_3 = X - n_1 - n_2$. Then,

$$\triangle_3 M(H, c_X^*) = \left[ \binom{X}{2} - \binom{X - n_1 - n_2}{2} \right] - \left[ \binom{N - X - 1}{2} - \binom{N - X - 1}{2} \right] \geqslant 0.$$

The equality holds only when $X < 2$, which is impossible. This yields a contrary result: The number of monochromatic edges increases when the number of red vertices increases. Hence, $F_{10}$ gives the minimum number of monochromatic edges when $X = n_1 + n_2$ instead.

Next, we will consider the last two cases of type B hypergraphs.

Case 4B: $n_2 \leqslant X < n_3$ and $H$ is a type B hypergraph.

We have that $c_X^*$ is in the form $F_3$ or $F_6$. We will show that $\triangle_3 M(H, c_X^*) < 0$ for $F_3$ and $\triangle_1 M(H, c_X^*) < 0$ for $F_6$. If $c_X^*$ is in the form $F_3$ with $x_1 = 0$, $x_2 = 0$ and $x_3 = X$, then

$$\triangle_3 M(H, c_X^*) = \left[ \binom{X}{2} - \binom{X}{2} \right] - \left[ \binom{N - X - 1}{2} - \binom{n_3 - X - 1}{2} \right] \leqslant 0.$$

Note that the inequality is equal when $N - X - 1 < 2$, which is impossible. Next, if $c_X^*$ is in the form $F_6$ with $x_1 = X - n_2$, $x_2 = n_2$ and $x_3 = 0$, then

$$\triangle_1 M(H, c_X^*) = \left[ \binom{X}{2} + \binom{n_1 + n_2 - X - 1}{2} \right] - \left[ \binom{N - X - 1}{2} + \binom{X - n_2}{2} \right].$$

We will show that $\triangle_1 M(H, c_X^*) < 0$ by Proposition 1. We have $X + (n_1 + n_2 - X - 1) = n_1 + n_2 - 1 \leqslant n_1 + n_3 - 1 = (N - X - 1) + (X - n_2)$, $X > X - n_2$ and $n_1 + n_2 - X - 1 < N - X - 1$. Since $2 \leqslant N - X - 1$, we have that $\triangle_1 M(H, c_X^*) < 0$.

Case 5B: $n_3 \leqslant X \leqslant \lfloor \frac{N}{2} \rfloor$ and $H$ is a type B hypergraph.

We have that $c_X^*$ is in the form $F_8$ with $x_1 = X - n_3$, $x_2 = 0$ and $x_3 = n_3$. Then,

$$\triangle_1 M(H, c_X^*) = \left[ \binom{X}{2} - \binom{X - n_3}{2} \right] - \left[ \binom{N - X - 1}{2} - \binom{n_1 - (X - n_3) - 1}{2} \right]$$
$$= \frac{n_1^2 + 2n_1 n_3 + 2X(N - n_1) - 3n_1 + 3N - N^2 - 4n_3}{2}.$$

Since we cannot apply any lemmas to $\triangle_1 M(H, c_X^*)$, we expand the binomial coefficient terms and see when $\triangle_1 M(H, c_X^*)$ is fewer than 0. Consequently, $\triangle_1 M(H, c_X^*) < 0$ if and only if

$$X < \left\lceil \frac{N^2 - 3N - n_1^2 - 2n_1 n_3 + 3n_1 + 4n_3}{2(N - n_1)} \right\rceil.$$

Write $X'$ for $\left\lceil \frac{N^2 - 3N - n_1^2 - 2n_1 n_3 + 3n_1 + 4n_3}{2(N - n_1)} \right\rceil$. We have that the $c_{X'}^*$ has the minimum number of monochromatic edges among all colorings in the form $F_8$, and we will show that $X' \leqslant \frac{N}{2}$.

$$X' = \left\lceil \frac{N^2 - 3N - n_1^2 - 2n_1 n_3 + 3n_1 + 4n_3}{2(N - n_1)} \right\rceil \leqslant \frac{N^2 - Nn_1 - 2n_3 + 2n_3}{2(N - n_1)} = \frac{N}{2}.$$

Hence, we have shown all the comparisons between colorings, and we can conclude that:

1. If $H$ is a type A hypergraph, then the coloring with $n_1 + n_2$ red vertices in the form $F_{10}$ has the minimum number of monochromatic edges.;
2. If $H$ is a type B hypergraph, then the coloring with $X'$ red vertices in the form $F_8$ has the minimum number of monochromatic edges.

We have already proved that those minimum colorings are the unique colorings that have the minimum number of monochromatic edges among colorings with the same, such as red, vertices. Furthermore, we have shown that $\triangle M(H, c_X^*)$ is fewer than zero when $X$ is fewer than $n_1 + n_2$ in a type A hypergraph and when $X$ is less than $X'$ in a type B hypergraph. Hence, those minimum colorings are the unique colorings that have the minimum number of monochromatic edges among all colorings.

*4.2. Hypergraphs with $n_1 < 3$ or $n_2 < 3$*

In this subsection, we will prove the remaining cases, which are unbalance complete tripartite hypergraphs with some classes smaller than 3. These cases are easy and straightforward but contain fuzzy details. First, we will consider an unbalanced complete tripartite 3-uniform hypergraphs with $n_1 \leqslant n_2 < n_3 \geqslant 3$. There are three possibilities for these hypergraphs:

Case i: $n_1 = n_2 = 1$ and $n_3 \geqslant 3$.

Since we have that the first two classes are smaller than 3, no edge can be contained in the first two classes. Thus,

$$M(H, c) = \binom{X}{3} + \binom{N - X}{3} - \binom{x_3}{3} - \binom{n_3 - x_3}{3}.$$

We have that $M(H, c) \geqslant 0$. Suppose that $M(H, c) = 0$. Since $X \geqslant x_3$ and $N - X \geqslant n_3 - x_3$, we have that $\binom{X}{3} = \binom{x_3}{3}$ and $\binom{N-X}{3} = \binom{n_3-x_3}{3}$. Since $N = n_1 + n_2 + n_3 = 1 + 1 + n_3 \geqslant 5$, at least three vertices are colored the same, such as red, i.e., $X \geqslant 3$. Then, $X = x_3$, which implies that $N - X = n_3 + 2 - X > n_3 - x_3$. Consequently, $3 > N - X = n_3 + 2 - x_3$, i.e., $n_3 = x_3$. This means that $c$ is a coloring such that the third class contains all red vertices and no blue vertex, and the first two classes are all blue. Hence, we have already determined the minimum coloring and showed that it is unique up to a permutation of colors and classes.

Case ii: $n_1 = 1, n_2 = 2$ and $n_3 \geqslant 3$.

Again, no edge can be contained in the first two classes. Thus,

$$M(H, c) = \binom{X}{3} + \binom{N - X}{3} - \binom{x_3}{3} - \binom{n_3 - x_3}{3}.$$

We will show that $M(H, c) \geqslant 1$. We have that $X \geqslant x_3$ and $N - X \geqslant n_3 - x_3$. Since $N = n_1 + n_2 + n_3 = 1 + 1 + n_3 \geqslant 6$, at least three vertices are colored the same, such as red, i.e., $X \geqslant 3$. If $X > x_3$, then $M(H, c) > 0$. Suppose that $X = x_3$. Then, $N - X = n_3 + 3 - x_3 > n_3 - x_3 \geqslant 3$. Hence, $M(H, c) > 0$. Next, suppose that $M(H, c) = 1$. If $X > x_3$, then

$$1 \leqslant \binom{X - 1}{2} \leqslant \binom{X}{3} - \binom{x_3}{3} \leqslant \binom{X}{3} + \binom{N - X}{3} - \binom{x_3}{3} - \binom{n_3 - x_3}{3} = M(H, c).$$

This implies that $X = 3$, and so, $N - X \geqslant 3$; moreover, $\binom{N-X}{3} = \binom{n_3-x_3}{3}$. Hence, $N - X = n_3 - x_3$. Now, we have $N - 3 = N - X = n_3 - x_3 = N - 3 - x_3$, i.e., $x_3 = 0$. This means $c$ is a coloring such that the third class contains no red vertices, and the first two classes are all red.

Suppose that $X = x_3$. Then,

$$1 = M(H, c) = \binom{X}{3} + \binom{N - X}{3} - \binom{x_3}{3} - \binom{n_3 - x_3}{3} = \binom{n_3 - x_3 + 3}{3} - \binom{n_3 - x_3}{3}.$$

It is only possible when $n_3 - x_3 = 0$ or when $c$ is a coloring such that the third class contains all red vertices and no blue vertex and the first two classes are all blue. Note that if we relabel the names of colors, then both colorings are the same. Hence, we have already determined the minimum coloring and showed that it is unique up to a permutation of colors and classes.

Case iii: $n_1 = 2, n_2 = 2$ and $n_3 \geqslant 3$.

Similarly, no edges can be contained in the first two classes. Thus,

$$M(H, c) = \binom{X}{3} + \binom{N - X}{3} - \binom{x_3}{3} - \binom{n_3 - x_3}{3}.$$

We will show that $M(H, c) \geqslant 4$. We have that $X \geqslant x_3$ and $N - X \geqslant n_3 - x_3$. If there is a color, say red, such that all vertices of that color are only in the third class, then, we have $X = x_3$ and $N - X = n_3 + 4 - x_3 \geqslant 4$. Thus,

$$\begin{aligned}
M(H, c) &= \binom{N - X}{3} - \binom{N - X - 4}{3} \\
&= \binom{N - X - 1}{2} + \binom{N - X - 2}{2} + \binom{N - X - 3}{2} + \binom{N - X - 4}{2} \\
&\geqslant \binom{3}{2} + \binom{2}{2} = 4.
\end{aligned}$$

The equality holds only when $N - X = 4$, i.e., $n_3 = x_3$. Hence, if $M(H, c) = 4$, then $c$ is a coloring such that the third class contains all red vertices but no blue vertex and the first two classes are all blue.

Suppose that there is no color such that all vertices of that color are only in the third class, i.e., $X > x_3$ and $N - X > n_3 - x_3$. Since $N = n_1 + n_2 + n_3 = 2 + 2 + n_3 \geqslant 7$, at least four vertices are colored the same, such as red, i.e., $X \geqslant 4$. If $X \geqslant 5$, then

$$M(H, c) \geqslant \binom{X}{3} - \binom{x_3}{3} \geqslant \binom{X}{3} - \binom{X - 1}{3} = \binom{X - 1}{2} \geqslant \binom{4}{2} = 6.$$

If $X = 4$, then $N - X \geqslant 3$ and

$$M(H, c) \geqslant \binom{X}{3} - \binom{x_3}{3} + 1 \geqslant \binom{X}{3} - \binom{X - 1}{3} + 1 = \binom{X - 1}{2} + 1 = 4.$$

The equality holds only when $N - X = 3$ and $x_3 = X - 1$, i.e., $N = 7$ and $x_3 = 3$. This means that $n_3 = 3$. Hence, if $M(H, c) = 4$, then $c$ is a coloring of $H$, which has seven vertices such that the third class is all red, the second class is all blue and the first class has one red and one blue vertex. This implies that when $n_3 = 3$, minimum colorings are not unique.

Finally, the last class is a hypergraph such that only the first class is smaller than 3.

Case iv: $n_1 < 3$ and $3 \leqslant n_2 \leqslant n_3$.

Fortunately, this case conforms to almost all cases in the previous subsection since we assume that $n_2 \geqslant 3$. However, there are two points that we use the fact that $n_1 \geqslant 3$. The first one is in the last part of Case 3 where we compare and determine which form of $F_2$ and $F_4$ has fewer monochromatic edges. Since we do not have that $n_1 \geqslant 3$, it is possible that both forms have the same number of monochromatic edges. This is not problematic because both of them have strictly more monochromatic edges than some coloring according to the next comparisons. The next case is in the last part of Case 5B where we compare and determine which form of $F_6$ and $F_8$ has fewer monochromatic edges. Again, since we do not have that $n_1 \geqslant 3$, we may have that $n_1 + n_3 - X < 3$, where $n_2 < n_3 \leqslant X < \dfrac{n_1 + n_2 + n_3}{2}$ and $n_1 + n_2 > n_3$. If this condition occurs, we will have that $F_6$ and $F_8$ have the same

number of monochromatic edges. Since $n_1 < 3$, the condition is possible only when $N$ is even and $n_1 < 3$ or $N$ is odd and $n_1 \leqslant 3$.

Suppose that $N$ is even. If $n_1 = 1$, then we have that $n_3 < n_1 + n_2 = n_2 + 1$. Since $n_2 < n_3$, there is no choice for $n_3$. If $n_1 = 2$, then we have that $n_3 < n_1 + n_2 = n_2 + 2$. Since $n_2 < n_3$, $n_3 = n_2 + 1$. However, for $N$ to be even, $n_2$ and $n_3$ must have different parities, which is impossible.

Suppose that $N$ is odd. Again, it is impossible for $n_1 = 1$. Since $n_1 < 3$, we have $n_1 = 2$. Similarly, $n_3 = n_2 + 1$ and $N = n_1 + n_2 + n_3 = 2 + n_2 + n_2 + 1 = 2(n_2 + 1) + 1$, which is odd. Consequently, the condition is possible only here and $F_6$ and $F_8$ have the same number of monochromatic edges. Hence, if $H$ is a hypergraph with $n_1 = 2$ and $3 \leqslant n_2 = n_3 - 1$, then there are only two colorings (each unique up to a permutation of colors and classes) that have a minimum number of monochromatics, which are colorings with $X = \lfloor \frac{N}{2} \rfloor$ in the form $F_6$ and $F_8$.

Now, we have determined the minimum colorings of all unbalanced complete tripartite 3-uniform hypergraphs.

### 5. Proof of Theorem 3

Assume that $k \geqslant 2$. Let $H$ be a balanced complete $k$-partite $(r+1)$-uniform hypergraph with $n \geqslant r + 1$ vertices in each class, and let $N = kn$. Let $c$ be a red/blue/green coloring of $H$ with the numbers of red, blue and green vertices of the $i$th class equal to $r_i, b_i$ and $g_i$, respectively, and let $R$, $B$ and $G$ be the total numbers of red, blue and green vertices, respectively.

Let $\triangle_{ii'} M(H, c, r, b)$ be the change in the number of monochromatic edges if a red vertex in the $i$th class is recolored into blue and a blue vertex in the $i'$th class is recolored into red. The definitions are similar for other color combinations. The process will be called *swapping*, which results in a new coloring, say $c'$. As a result of the process, the number of red vertices in the $i$th class decreases by 1, and the number of red vertices in the $i'$th class increases by 1, while the total number of red vertices remains the same. In other words, the coloring $c'$ has $r_i - 1$ and $r_{i'} + 1$ red vertices in the $i$th and $i'$th classes, respectively. Likewise, the coloring $c'$ has $b_i + 1$ and $b_{i'} - 1$ blue vertices in the $i$th and $i'$th classes, respectively.

We can compute $\triangle_{ii'} M(H, c, r, b)$ by comparing the numbers of monochromatic edges containing those vertices that underwent swapping before and after the swapping process. Thus,

$$\triangle_{ii'} M(H, c, r, b) = \left[ \binom{B-1}{r} - \binom{b_i}{r} + \binom{R-1}{r} - \binom{r_{i'}}{r} \right]$$
$$- \left[ \binom{R-1}{r} - \binom{r_i - 1}{r} + \binom{B-1}{r} - \binom{b_{i'} - 1}{r} \right]$$
$$= \left[ \binom{r_i - 1}{r} + \binom{b_{i'} - 1}{r} \right] - \left[ \binom{b_i}{r} + \binom{r_{i'}}{r} \right].$$

A *successful swapping* is a swapping in such a way that the number of monochromatic edges is reduced, i.e., $\triangle_{ii'} M(H, c, r, b) < 0$. Note that if $0 < r_i < n, < b_i < n, r_{i'} = n - 1$ and $b_{i'} = 1$, then

$$\triangle_{ii'} M(H, c, r, b) = \left[ \binom{r_i - 1}{r} + \binom{b_{i'} - 1}{r} \right] - \left[ \binom{b_i}{r} + \binom{r_{i'}}{r} \right]$$
$$= \left[ \binom{r_i - 1}{r} + \binom{1 - 1}{r} \right] - \left[ \binom{b_i}{r} + \binom{n - 1}{r} \right].$$

Since $r_i < n$, $0 < b_i$ and $n - 1 \geqslant r$, we have that $\triangle_{ii'} M(H, c, r, b) < 0$. This implies that a swapping resulting in fewer polychromatic classes is always successful.

**Lemma 3.** *If $\triangle_{ii'} M(H, c, r, b) \leqslant 0$, then $\triangle_{ii'} M(H, c', r, b) \leqslant 0$.*

The proof of this lemma is similar to that of Lemma 1. Lemma 3 means that if swapping can be carried out without increasing the number of monochromatic edges, another swapping in the same direction will be successful (if there is a red and blue vertices to be swapped). The process of successful swappings will terminate when the $i$th class has no red vertex or the $i'$th class has no blue vertex.

**Lemma 4.** *If $\triangle_{ii'} M(H, c, r, b) \geqslant 0$, then $\triangle_{ii'} M(H, c, b, r) \leqslant 0$.*

The proof of this lemma is similar to that of Lemma 2. However, in contrast to Lemma 2, it is possible that the equality holds. Note that if $c$ contains two classes, the $i$th and $i'$th, such that both of them contain at least one red vertex and one blue vertex, then there are two directions of swapping as follows:

1.　Swapping a red vertex of the $i$th class with a blue vertex of the $i'$th class;
2.　Swapping a blue vertex of the $i$th class with a red vertex of the $i'$th class.

By Lemma 4, one of the two directions can be achieved without increasing the number of monochromatic edges. Moreover, by Lemma 3, we can continue swapping in the same direction until the $i$th class has no red vertex or the $i'$th class has no blue vertex, and the number of monochromatic edges does not increase. Note that we obtain the same result when considering swapping relating to other color combinations. Hence, the coloring with the minimum number of monochromatic edges among colorings with a constant number of red, blue and green vertices is the coloring such that, for any two classes, they must have at most one color of vertices to be in common. We will list all these forms in Table 11.

**Table 11.** The five canonical forms for three colors.

| Canonical Forms | Descriptions |
| :---: | :---: |
| $F_1$ | The first class contains three colors while the other classes are monochromatic. |
| $F_2$ | The first class contains a pair of colors while the other classes are monochromatic. |
| $F_3$ | The first and second classes contain different pairs colors while the other classes are monochromatic. |
| $F_4$ | The first, second and third classes contain different pairs colors while the other classes are monochromatic. |
| $F_5$ | All classes are monochromatic. |

The first column illustrates the list of five canonical forms, and the second column describes the colors of vertices in each class.

Suppose that $r_i \leqslant b_{i'}$. Let $\triangle_{ii'} M_T(H, c, r, b)$ be the change in the number of monochromatic edges if all red vertices in the $i$th class are recolored into blue and $r_i$ blue vertices in the $i'$th class are recolored into red. The process will be called *total swapping*, which results in a new coloring, say $c'$. We can compute $\triangle_{ii'} M_T(H, c, r, b)$ by summing the change in the number of monochromatic edges of the swappings. Thus,

$$
\begin{aligned}
\triangle_{ii'} M_T(H, c, r, b) &= \sum_{k=0}^{r_i - 1} \left[ \binom{r_i - 1 - k}{r} + \binom{b_{i'} - 1 - k}{r} \right] - \left[ \binom{b_i + k}{r} + \binom{r_{i'} + k}{r} \right] \\
&= \left[ \binom{0}{r} + \binom{1}{r} + \cdots + \binom{r_i - 1}{r} \right] + \left[ \binom{b_{i'} - r_i}{r} + \binom{b_{i'} - r_i + 1}{r} + \cdots + \binom{b_{i'} - 1}{r} \right] \\
&\quad - \left[ \binom{b_i}{r} + \binom{b_i + 1}{r} + \cdots + \binom{b_i + r_i - 1}{r} \right] - \left[ \binom{r_{i'}}{r} + \binom{r_{i'} + 1}{r} + \cdots + \binom{r_{i'} + r_i - 1}{r} \right].
\end{aligned}
$$

A *successful total swapping* is a total swapping in such a way that the number of monochromatic edges is reduced.

**Lemma 5.** *Suppose that* $1 \leqslant r_i \leqslant b_{i'}$. *Then,* $\triangle_{ii'} M_T(H, c, r, b) \leqslant 0$ *if c satisfies at least one of the following conditions:*

1.  $b_{i'} < r_i + b_i$;
2.  $b_{i'} < r_i + r_{i'}$.

**Proof.** Suppose that $1 \leqslant r_i \leqslant b_{i'}$. If $b_{i'} < r_i + b_i$, then

$$\triangle_{ii'} M_T(H, c, r, b)$$

$$\leqslant \left[ \binom{0}{r} + \binom{1}{r} + \cdots + \binom{r_i - 1}{r} \right] - \left[ \binom{r_{i'}}{r} + \binom{r_{i'} + 1}{r} + \cdots + \binom{r_{i'} + r_i - 1}{r} \right] \leqslant 0.$$

The equality holds only when $b_i + r_i - 1 < r$ and $r_{i'} + r_i - 1 < r$, i.e., $b_i + r_i < r + 1$ and $r_i + r_{i'} < r + 1$. If $b_{i'} < r_i + r_{i'}$, then

$$\triangle_{ii'} M_T(H, c, r, b) \leqslant \left[ \binom{0}{r} + \binom{1}{r} + \cdots + \binom{r_i - 1}{r} \right] - \left[ \binom{b_i}{r} + \binom{b_i + 1}{r} + \cdots + \binom{b_i + r_i - 1}{r} \right]$$

$$\leqslant 0.$$

The equality holds only when $b_i + r_i - 1 < r$ and $r_{i'} + r_i - 1 < r$, i.e., $b_i + r_i < r + 1$ and $r_i + r_{i'} < r + 1$. $\square$

Note that if we consider a class with only two colors—say $r_i + b_i = n \geqslant r + 1$—then $\triangle_{ii'} M_T(H, c, r, b)$ is strictly less than zero in both cases. We will use Lemma 5 to obtain more information from the canonical forms. From this point, the term *quadruple* refers to a collection of four values that are the numbers of vertices of any pair of colors in any pair of classes, e.g., $(r_i, b_i, r_{i'}, b_{i'})$. Consequently, in each coloring in the forms any canonical forms, all quadruples contain at least one zero. Suppose that $c$ contains the $i$th and $i'$th classes such that they have a color in common, but the $i$th class contains a color that the $i'$th class does not; for example, we focus on $(r_i \neq 0, b_i \neq 0, r_{i'} = 0, b_{i'} \neq 0)$. If $c$ is a minimum coloring with $r_i + b_i = n$, then we can conclude from Lemma 5 that

1.  If $r_i \leqslant b_{i'}$, then $b_{i'} \geqslant r_i + b_i$ and $b_{i'} \geqslant r_i + r_{i'} = r_i$;
2.  If $r_i \geqslant b_{i'}$, then $r_i \geqslant b_i + b_{i'}$ and $r_i \geqslant r_{i'} + b_{i'} = b_{i'}$.

We will call these *quadruple conditions*.

Next, we will focus on the possibility of $F_4$. Suppose that $c$ is the coloring that has the minimum number of monochromatic edges, which is in the form $F_4$ such that, without loss of generality, the first class has $g_1 \neq 0$ green and $r_1 \neq 0$ red vertices, the second class has $g_2 \neq 0$ green and $b_2 \neq 0$ blue vertices and the third class has $r_3 \neq 0$ red and $b_3 \neq 0$ blue vertices, where $g_1$ is the maximum among those values. We will apply quadruple conditions to the quadruple $(g_1, b_1 = 0, g_2, b_2)$. Since $g_1 \geqslant b_2$ and $g_2 + b_2 = n$, $n = g_2 + b_2 \leqslant g_1 = n - r_1 < n$, which is a contradiction. Hence, a minimum coloring cannot be in the form $F_4$, and this form will be out of our interest. Consequently, we will search for the minimum colorings only from $F_1, F_2, F_3$ and $F_5$.

We will divide this into two cases upon the remainder of the number $k$ of vertex classes divided by three. For simplicity, we define a *type i* hypergraph to be a hypergraph with $k \equiv i \pmod 3$ classes for $i = 0, 1, 2$. We will consider a type 0 hypergraph first.

Case 0: $H$ is a type 0 hypergraph.

Since the number of classes is divisible by the number of colors, it is conducted using Proposition 3.

Case 1: $H$ is a type 1 hypergraph.

Case 1.1: The coloring $c$ is in the form $F_3$.

In this case, our aim is to show that all colorings in the form $F_3$ have more monochromatic edges than some coloring in the form $F_2$. We have that $c$ has two classes that are polychromatic—say the first class contains green and red vertices and the second class contains green and blue vertices—while the rest of the classes are monochromatic. The

number of vertices of each color and the number of monochromatic classes of each color are shown in Tables 12 and 13.

**Table 12.** The number of vertices of each color for the first two classes.

| Polychromatic Classes | Number of Red Vertices | Number of Blue Vertices | Number of Green Vertices |
|---|---|---|---|
| 1 | $r_1 \neq 0$ | 0 | $g_1 \neq 0$ |
| 2 | 0 | $b_2 \neq 0$ | $g_2 \neq 0$ |

**Table 13.** The number of monochromatic classes of each color.

| Number of Red Classes | Number of Blue Classes | Number of Green Classes |
|---|---|---|
| $k_r$ | $k_b$ | $k_g$ |

We will show that the coloring where $k_r$, $k_b$ and $k_g$ are as equal as possible has a smaller number of monochromatic edges than the coloring which is not. Without loss of generality, suppose that $c$ has $k_g$ green classes and $k_r$ red classes such that $k_g - k_r \geqslant 2$. We will recolor all vertices in a green class into red and obtain a new coloring $c'$. Thus,

$$M(H,c) = \binom{nk_r + r_1}{r+1} + \binom{nk_b + b_2}{r+1} + \binom{nk_g + g_1 + g_2}{r+1}$$
$$- \binom{r_1}{r+1} - \binom{b_2}{r+1} - \binom{g_1}{r+1} - \binom{g_2}{r+1} - (k-2)\binom{n}{r+1}$$

and

$$M(H,c') = \binom{n(k_r+1) + r_1}{r+1} + \binom{nk_b + b_2}{r+1} + \binom{n(k_g-1) + g_1 + g_2}{r+1}$$
$$- \binom{r_1}{r+1} - \binom{b_2}{r+1} - \binom{g_1}{r+1} - \binom{g_2}{r+1} - (k-2)\binom{n}{r+1}.$$

Then,

$$M(H,c') - M(H,c) = \left[ \binom{n(k_r+1) + r_1}{r+1} + \binom{n(k_g-1) + g_1 + g_2}{r+1} \right]$$
$$- \left[ \binom{nk_r + r_1}{r+1} + \binom{nk_g + g_1 + g_2}{r+1} \right].$$

We will show that $M(H,c') - M(H,c) < 0$ by Proposition 1. We have $[n(k_r+1) + r_1] + [n(k_g-1) + g_1 + g_2] = [nk_r + r_1] + [nk_g + g_1 + g_2]$, $nk_r + r_1 < n(k_r+1) + r_1 \leqslant n(k_g - 1) + r_1 < nk_g + g_1 + g_2$ and $nk_r + r_1 < n(k_r+1) + g_1 + g_2 \leqslant n(k_g-1) + g_1 + g_2 < nk_g + g_1 + g_2$. Since $nk_g + g_1 + g_2 > r + 1$, we have that $M(H,c') - M(H,c) < 0$. Note that we only use the fact that $r_1 < n$ and $g_1 + g_2 > 0$.

Since $(k-2) \equiv 2 \pmod 3$, colorings such that $k_r$, $k_b$ and $k_g$ are as equal as possible must have one of these conditions:

1. $k_r + 1 = k_b = k_g$;
2. $k_r = k_b + 1 = k_g$;
3. $k_r = k_b = k_g + 1$.

We will show that a coloring with the last condition has more monochromatic edges than a coloring with either of the first two conditions (with the same polychromatic classes). Suppose that $c$ is a coloring such that $k_r = k_b = k_g + 1$. We consider $(g_1, r_1, g_2, 0)$ of $c$. Since $g_1 + r_1 = n$, if $g_2 \geqslant r_1$, then $g_2 \geqslant r_1 + g_1$. This leads to a contradiction since $n = r_1 + g_1 \leqslant g_2 = n - b_2 < n$. Hence, $g_2 < r_1$ and consequently $g_1 + g_2 \leqslant r_1$ by quadruple conditions. Next, we will increase the number of green classes. Without loss of

generality, we will recolor all vertices in a red class into green and obtain a new coloring $c'$ with Condition (1). Then,

$$
\begin{aligned}
M(H,c) = {} & \binom{nk_r + r_1}{r+1} + \binom{nk_b + b_2}{r+1} + \binom{nk_g + g_1 + g_2}{r+1} \\
& - \binom{r_1}{r+1} - \binom{b_2}{r+1} - \binom{g_1}{r+1} - \binom{g_2}{r+1} - (k-2)\binom{n}{r+1}
\end{aligned}
$$

and

$$
\begin{aligned}
M(H,c') = {} & \binom{n(k_r - 1) + r_1}{r+1} + \binom{nk_b + b_2}{r+1} + \binom{n(k_g + 1) + g_1 + g_2}{r+1} \\
& - \binom{r_1}{r+1} - \binom{b_2}{r+1} - \binom{g_1}{r+1} - \binom{g_2}{r+1} - (k-2)\binom{n}{r+1}.
\end{aligned}
$$

Then,

$$
\begin{aligned}
M(H,c') - M(H,c) = {} & \left[ \binom{n(k_r - 1) + r_1}{r+1} + \binom{n(k_g + 1) + g_1 + g_2}{r+1} \right] \\
& - \left[ \binom{nk_r + r_1}{r+1} + \binom{nk_g + g_1 + g_2}{r+1} \right].
\end{aligned}
$$

We will show that $M(H,c') - M(H,c) < 0$ by Proposition 1. We have $[n(k_r - 1) + r_1] + [n(k_g + 1) + g_1 + g_2] = [nk_r + r_1] + [nk_g + g_1 + g_2]$. Since $g_1 + g_2 \leqslant r_1$, then $nk_g + g_1 + g_2 \leqslant nk_g + r_1 = n(k_r - 1) + r_1 < nk_r + r_1$ and $nk_g + g_1 + g_2 < n(k_g + 1) + g_1 + g_2 \leqslant n(k_g + 1) + r_1 = nk_r + r_1$. Since $nk_g + g_1 + g_2 > r + 1$, we have that $M(H,c') - M(H,c) < 0$.

Finally, we will consider a coloring with condition (1) or (2). By symmetry, assume that $c$ is a coloring with $k_r + 1 = k_b = k_g$. We will recolor a green vertex in the first class of $c$ into red. Then,

$$
\triangle_1 M(H,c) = \left[ \binom{nk_r + r_1}{r} - \binom{nk_g + g_1 + g_2 - 1}{r} \right] + \left[ \binom{g_1 - 1}{r} - \binom{r_1}{r} \right].
$$

We have $nk_r + r_1 = n(k_g - 1) + r_1 < nk_g < nk_g + g_1 + g_2 - 1$ and $g_1 - 1 < g_1 + g_2 \leqslant r_1$. Since $nk_g + g_1 + g_2 - 1 > r$, we have that $\triangle_1 M(H,c) < 0$. This is true for any arbitrary value of $r_1 > 0$. Hence, we will recolor a green vertex into red until the first class is a red class, which is a coloring in the form $F_2$, and obtain fewer monochromatic edges. Consequently, we can conclude that $c$ has more monochromatic edges than a coloring in the form $F_2$.

Case 1.2: The coloring $c$ is in the form $F_5$.

In this case, our aim is to show that all colorings in the form $F_5$ have more monochromatic edges than some coloring in the form $F_2$. We have that all classes of $c$ are monochromatic with $k_r$, $k_b$ and $k_g$ red, blue and green classes, respectively. We can show that the coloring where $k_r$, $k_b$ and $k_g$ are as equal as possible has a smaller number of monochromatic edges similarly as in Case 1.1. Since $k \equiv 1 \pmod 3$, colorings where $k_r$, $k_b$ and $k_g$ are as equal as possible must have one of these conditions:

1. $k_r - 1 = k_b = k_g$;
2. $k_r = k_b - 1 = k_g$;
3. $k_r = k_b = k_g - 1$.

By symmetry, suppose that $c$ is a coloring such that $k_r - 1 = k_b = k_g$. We will show that if a red vertex in a red class, say the first class, is recolored into green, the number of

monochromatic edges will decrease. The new coloring is not in the form $F_5$ but in the form $F_2$ instead. Then,

$$\triangle_1 M(H, c) = \left[ \binom{nk_g}{r} + \binom{n-1}{r} \right] - \left[ \binom{nk_r - 1}{r} + \binom{0}{r} \right].$$

we will show that $\triangle_1 M(H, c) < 0$ by Proposition 1. We have $nk_g + (n-1) = n(k_g + 1) - 1 = (nk_r - 1) + 0$ and $0 < n - 1 < nk_g < nk_r - 1$. Since $nk_r - 1 > r$, we have that $\triangle_1 M(H, c) < 0$. Consequently, we can conclude that $c$ has more monochromatic edges than a coloring in the form $F_2$.

Case 1.3: The coloring $c$ is in the form $F_2$.

In this case, our aim is to show that all colorings in the form $F_2$ have more monochromatic edges than some coloring in the form $F_1$. We have that $c$ has a class that is polychromatic. Without loss of generality, say the first class contains red and blue vertices while the rest classes are monochromatic. Note that the number of vertices of each color and the number of monochromatic classes of each color are shown in Tables 14 and 15.

**Table 14.** The number of vertices of each color for the first class.

| Polychromatic Classes | Number of Red Vertices | Number of Blue Vertices | Number of Green Vertices |
|---|---|---|---|
| 1 | $r_1 \neq 0$ | $b_1 \neq 0$ | 0 |

**Table 15.** The number of monochromatic classes of each color.

| Number of Red Classes | Number of Blue Classes | Number of Green Classes |
|---|---|---|
| $k_r$ | $k_b$ | $k_g$ |

We can show that the coloring where $k_r$, $k_b$ and $k_g$ are as equal as possible has a smaller number of monochromatic edges, similarly to Case 1.1. Hence, we will focus on a coloring where $k_r$, $k_b$ and $k_g$ are as equal as possible. Since $(k-1) \equiv 0 \pmod 3$, colorings where $k_r$, $k_b$ and $k_g$ are as equal as possible must have $k_r = k_b = k_g$.

Suppose that $c$ is coloring such that $k_r = k_b = k_g$. By symmetry, suppose that $r_1 > 1$. We will show that if a red vertex in the first class is recolored into green, the number of monochromatic edges will decrease. The new coloring is not in the form $F_2$ but in the form $F_1$ instead. Then,

$$\triangle_1 M(H, c) = \left[ \binom{nk_g}{r} - \binom{0}{r} \right] - \left[ \binom{nk_r + r_1 - 1}{r} - \binom{r_1 - 1}{r} \right]$$
$$= \left[ \binom{nk_g}{r} + \binom{r_1 - 1}{r} \right] - \left[ \binom{nk_r + r_1 - 1}{r} + \binom{0}{r} \right].$$

We will show that $\triangle_1 M(H, c) < 0$ by Proposition 1. We have $nk_g + (r_1 - 1) = (nk_r + r_1 - 1) + 0$ and $0 < r_1 - 1 < nk_g < nk_r + r_1 - 1$. Since $nk_r + r_1 - 1 > r$, we have that $\triangle_1 M(H, c) < 0$. Consequently, we can conclude that $c$ has more monochromatic edges than coloring in the form $F_1$.

Case 1.4: The coloring $c$ is in the form $F_1$.

In this case, our aim is to show that coloring in the form $F_1$ such that the numbers of red, blue and green vertices are as equal as possible has the minimum number of monochromatic edges. We have that $c$ has a class that is polychromatic. Without loss of generality, say the first class contains red, blue and green vertices, while the rest classes are monochromatic. Note that the number of vertices of each color and the number of monochromatic classes of each color are shown in Tables 16 and 17.

**Table 16.** The number of vertices of each color for the first class.

| Polychromatic Classes | Number of Red Vertices | Number of Blue Vertices | Number of Green Vertices |
|:---:|:---:|:---:|:---:|
| 1 | $r_1 \neq 0$ | $b_1 \neq 0$ | $g_1 \neq 0$ |

**Table 17.** The number of monochromatic classes of each color.

| Number of Red Classes | Number of Blue Classes | Number of Green Classes |
|:---:|:---:|:---:|
| $k_r$ | $k_b$ | $k_g$ |

We can show that the coloring where $k_r$, $k_b$ and $k_g$ are as equal as possible has a smaller number of monochromatic edges, as in Case 1.1. Since $(k-1) \equiv 0 \pmod 3$, colorings where $k_r$, $k_b$ and $k_g$ are as equal as possible must have $k_r = k_b = k_g$.

Suppose that $c$ is a coloring such that $k_r = k_b = k_g$. We will show that if we recolor the vertices in such a way that $r_1$, $b_1$ and $g_1$ are as equal as possible, then the number of monochromatic edges will decrease. Without loss of generality, assume that $r_1 - g_1 \geqslant 2$. We will show that if a red vertex in the first class is recolored into green, the number of monochromatic edges will decrease. Then,

$$\triangle_1 M(H,c) = \left[ \binom{nk_g + g_1}{r} - \binom{g_1}{r} \right] - \left[ \binom{nk_r + r_1 - 1}{r} - \binom{r_1 - 1}{r} \right]$$
$$= \left[ \binom{nk_g + g_1}{r} + \binom{r_1 - 1}{r} \right] - \left[ \binom{nk_r + r_1 - 1}{r} + \binom{g_1}{r} \right].$$

We will show that $\triangle_1 M(H,c) < 0$ by Proposition 1. We have $(nk_g + g_1) + (r_1 - 1) = (nk_r + r_1 - 1) + g_1$ and $g_1 < r_1 - 1 < nk_g + g_1 < nk_r + r_1 - 1$. Since $nk_r + r_1 - 1 > r$, we have that $\triangle_1 M(H,c) < 0$. Consequently, we can conclude that $c$ is not a minimum coloring.

To sum up, we have proven that all colorings in the forms $F_2$, $F_3$ and $F_5$ have strictly more monochromatic edges than some coloring in the form $F_1$. Moreover, a coloring in the form $F_1$ where the numbers of red, blue and green vertices in the first class are not as equal as possible has strictly more monochromatic edges than the coloring $c^*$ where the numbers of red, blue and green classes are as equal as possible and the number of red, blue and green vertices in the first class are as equal as possible. Hence, this is the minimum coloring, and it is unique up to a permutation of colors and classes.

## 6. Concluding Remarks

In this paper, we considered 2-colorings of balanced complete $k$-partite $r$-uniform hypergraphs and determined which one has the minimum number of monochromatic edges. The proof may give a clue for further generalization to an unbalanced hypergraph with arbitrary sizes of classes. We observed that the minimum coloring can only be in certain forms, which are called canonical forms of the colorings. We studied the canonical forms of 2-colorings of unbalanced complete tripartite 3-uniform hypergraphs. Finally, we continued to determine the extermal 3-coloring of balanced complete $k$-partite $r$-uniform hypergraphs when $k \equiv 0, 1 \mod 3$.

In Theorem 2, we determined the minimum coloring for unbalanced 3-uniform hypergraphs. In the proof, almost all comparisons have been completed without expanding the binomial coefficient terms, and they also hold if we try to generalize the proof to $r$-uniform hypergraphs with arbitrary $r \leqslant n_1$. However, in the case when the largest class is smaller than the sum of two smaller classes and the number of red vertices is greater than the largest class, it cannot be completed without expanding those binomial coefficient terms where, in this case, we expand with $r + 1 = 3$ as the lower index. Hence, the proof works only for 3-uniform hypergraphs. We believe that the minimum coloring for $r$-uniform hypergraphs with arbitrary $r \leqslant n_1$ differs from those in Theorem 2.

Another generalized case is unbalanced complete hypergraphs with several vertex classes ($k > 3$). The problem seems to be much more complicated because Theorem 2 demonstrates that the extremal coloring varies depending on the relationship among the sizes of the vertex classes.

**Problem 2.** *What is the minimum 2-coloring of an unbalanced complete k-partite r-uniform hypergraph?*

We have determined the extermal 3-coloring of balanced complete $k$-partite $r$-uniform hypergraphs only for $k \equiv 0, 1 \mod 3$.

**Problem 3.** *What is the minimum 3-coloring of a balanced complete k-partite r-uniform hypergraph where $k \equiv 2 \mod 3$?*

For $k \equiv 2 \mod 3$, if we apply the same ideas as in the proof of Theorem 3, we can conclude that the minimum coloring is in the form $F_3$ instead of $F_1$. However, the comparisons between colorings in the form $F_3$ are rather challenging, and we believe that they require some further comparison tools. One might expect the minimum coloring to have approximately $\frac{V(H)}{3}$ vertices in each color as in Theorem 3. However, this is not the case, for example, when the number of vertices in each class is much greater than $k$.

For $m > 3$, we have only studied some trivial cases for $m$-colorings of the hypergraphs in Section 2.6.

**Problem 4.** *What is the minimum m-coloring of a balanced complete k-partite r-uniform hypergraph?*

This is a generalization of the hypergraphs in Problem 3, which is extremely complex as we believe that the minimum coloring varies depending on the relationship between the number of colors and the number of classes. However, the proof of Theorem 3 might be useful to determine the minimum $m$-coloring of balanced complete $k$-partite $r$-uniform hypergraphs with $k \equiv 1 \mod m$, and we speculate that it would be in the form similar to the coloring in the form $F_1$.

Finally, there is another natural definition of $k$-partite hypergraphs where each edge is an $r$-subset containing vertices from different classes.

**Problem 5.** *With the above definition of k-partite hypergraphs, what is the minimum coloring of a balanced complete k-partite r-uniform hypergraph?*

To sum up, we have studied the coloring of several types of complete multipartite hypergraphs from the simple ones to the generalized complex cases. This problem arose from the idea to match triples of students according to their genders and classes. The results are the minimum colorings of balanced complete $k$-partite $r$-uniform hypergraphs, unbalanced complete tripartite 3-uniform hypergraphs and balanced complete $k$-partite (for $k \equiv 0, 1 \mod 3$) $r$-uniform hypergraphs. In this paper, we constructed several propositions about the properties of binomial coefficients and used them as the main tools to prove each theorem. However, some theorems required further lemmas and pattern categorization (canonical forms) to be proven, and some also led to more generalized interesting problems.

**Author Contributions:** Conceptualization, T.K. and B.S.; Project administration, T.K.; Writing—original draft, B.S.; Writing—review & editing, T.K. All authors have read and agreed to the published version of the manuscript.

**Funding:** This research received no external funding.

**Conflicts of Interest:** The authors declare no conflict of interest.

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
