# Peer review of "Monochromatic Edges in Complete Multipartite Hypergraphs"

_mathematics, doi:10.3390/math10132353_

Round 1

Reviewer 1 Report

In the manuscript, the authors study the number of monochromatic edges of balanced complete k-partite runiform hypergraphs.

The results appear to be correct, although the length of the manuscript is considerable. The manuscript is well written, and the results and proofs are presented with clarity and technical rigor. Thus, I recommend the publication of this paper.

I suggest the authors to present the manuscript in the journal format, including the bibliographic references.

Author Response

Point 1 In the manuscript, the authors study the number of monochromatic edges of balanced complete k-partite r-uniform hypergraphs. The results appear to be correct, although the length of the manuscript is considerable. The manuscript is well written, and the results and proofs are presented with clarity and technical rigor. Thus, I recommend the publication of this paper. I suggest the authors present the manuscript in the journal format, including the bibliographic references.

Response 1: Done

Reviewer 2 Report

In this paper, 2-colorings of balanced complete k-partite r-uniform hypergraphs are studied.

The paper is well-written and consistent, the results are carefully proved. Since, graph coloring is a popular topic, I consider the paper of interest to readers. However, I believe that the literature review should be extended.

Author Response

Point 1 In this paper, 2-colorings of balanced complete k-partite r-uniform hypergraphs are studied. The paper is well-written and consistent, the results are carefully proved. Since graph coloring is a popular topic, I consider the paper of interest to readers.However, I believe that the literature review should be extended.

Response 1: We have revised our introduction part and added more relevant citations and references.

Reviewer 3 Report

Please see the attached review report.

Author Response

Point 1 Page 1: There should be a comma before and after “which is a popular topic.”

Response 1: Done

Point 2 Page 3: “we compares” should be “we compare.”

Response 2: Done

Point 3 Theorems start with 2 (1 is used for Problem 1). This is confusing. Each problem, lemma, claim, theorem, etc. should be numbered separately from 1.

Response 3: Done

Point 4 Proofs of theorems should be moved to the appendix instead of making each proof a separate section.

Response 4: We think that the proofs of theorems should not be moved due to the continuity of the paper. The arrangement of the proofs shows how each theorem is extended and generalized from the previous one which is the main content of the paper.

Point 5 Preliminaries should be right after the introduction, not after stating the theorems.

Response 5: For the continuity of the introduction part, we would like to introduce how the problem arises and then smoothly introduce the key results (apart from trivial ones) of this paper. Moreover, we change the name of the second section from “Preliminaries” into “Counting monochromatic edges” which would better suit the section. This section is designed to introduce helping propositions, counting methods, and also some trivial cases. 

Point 6 Proofs of theorems state further lemmas and their proofs. Lemmas should be stated

and proved before stating each theorem if it has to be used in its proof.

Response 6: Again for the continuity, we would like to place the lemmas and their proofs as a part of the proof of the main theorem. This emphasizes how the theorem differs from the previous one and needs a much more generalized concept to cope with. Again, if the lemmas are proved beforehands, they might confuse readers as they are not required for the proofs of some simpler theorems.

Point 7 Some sections repeat the title “Proof of Theorem X” right in the first paragraph. This

is redundant and confusing.

Response 7: We have already removed the redundant “Proof of Theorem X” at the beginning of each proof section.

Point 8 The paper is missing a conclusions section. “Concluding remarks” section is just like

another section in the paper, with further problems introduced and their solutions explained. Furthermore, the authors are referring to theorems and some numbered cases (like Case 5B). However, the conclusions section should be self-contained, the author doesn’t immediately know what case 5B represents.

Response 8: We think that we conclude all of the theorems and the results that we have done in this paper in this section. Further open problems are included to show that each of the theorems can be further generalized and emphasize that possibility. However, to conclude all things, we have added a paragraph at the end of this section to summarize the paper. Moreover, we make this section self-contained by not mentioning previous cases only by its abbreviation.

Point 9 Theorems are simply stating how some special cases would be solved. They don’t really rise to the level of a theorem. Get rid of the formality of theorems, just study them as

separate special cases and describe their solutions.

Response 9: We believe that the results deserve to be called theorems. Moreover, stating them as special cases might make it difficult to read. We would like to introduce our theorem to include all patterns of each type of the hypergraph.

Point 10 The literature review section is rather sparse. I would think there would be many more relevant papers to cite on this important subject. The authors should be able to cite

more than 11 references.

Response 10: We have revised our introduction part and added more relevant citations and references.